# DNAJB6 mutants display toxic gain of function through unregulated interaction with Hsp70 chaperones

Meital Abayev-Avraham[1], Yehuda Salzberg[2], Dar Gliksberg[1], Meital Oren-Suissa [2] & Rina Rosenzweig [1] ✉

Molecular chaperones are essential cellular components that aid in protein folding and preventing the abnormal aggregation of disease-associated proteins. Mutations in one such chaperone, DNAJB6, were identified in patients with LGMDD1, a dominant autosomal disorder characterized by myofibrillar degeneration and accumulations of aggregated protein within myocytes. The molecular mechanisms through which such mutations cause this dysfunction, however, are not well understood. Here we employ a combination of solution NMR and biochemical assays to investigate the structural and functional changes in LGMDD1 mutants of DNAJB6. Surprisingly, we find that DNAJB6 disease mutants show no reduction in their aggregation-prevention activity in vitro, and instead differ structurally from the WT protein, affecting their interaction with Hsp70 chaperones. While WT DNAJB6 contains a helical element regulating its ability to bind and activate Hsp70, in LGMDD1 disease mutants this regulation is disrupted. These variants can thus recruit and hyperactivate Hsp70 chaperones in an unregulated manner, depleting Hsp70 levels in myocytes, and resulting in the disruption of proteostasis. Interfering with DNAJB6-Hsp70 binding, however, reverses the disease phenotype, suggesting future therapeutic avenues for LGMDD1.

Molecular chaperones are a diverse group of proteins that play a crucial role in protecting cells from the dangers of protein misfolding and aggregation—phenomena that have been implicated in a host of debilitating human conditions, such as Alzheimer's, Parkinson's and Huntington's diseases, as well as many neuromuscular disorders and myopathies[1–6]. Chaperones perform this protective function by facilitating proper protein folding and assembly, refolding misfolded proteins to prevent their aggregation, and delivering damaged proteins for disposal[7–10].

With chaperones playing such an important role in protein homeostasis, it is therefore not surprising that failure of these cellular chaperone machinery can be highly detrimental, culminating in a wide range of pathologies. Limb girdle muscular dystrophy type 1D (LGMDD1) is an example of such a condition, caused by dominant

mutations to the molecular chaperone DNAJB6[6,11,12]. This incurable autosomal dominant muscle disorder is characterized by myofibrillar degeneration, autophagic vacuolation, and aggregation of myofibrillar proteins[11–14].

To date, 16 pathogenic mutations have been identified in DNAJB6[13,15,16]; however, the mechanism by which these mutations affect the structure and function of the chaperone, leading to LGMDD1 disease, is not fully understood.

Structurally, like all JDPs, DNAJB6 contains a conserved alpha-helical, N-terminal J-domain (Fig. 1a, b light gray) which, through direct interaction, stimulates Hsp70 ATPase activity, thereby activating the machinery[17]. The J-domain is followed by a glycine/phenylalanine (GF) rich region, which harbors the majority of LGMDD1 mutations[13,15,16], and has been suggested to participate both in client recognition and

[1]Department of Chemical and Structural Biology, Weizmann Institute of Science, Rehovot 761000, Israel. [2]Department of Brain Sciences, Weizmann Institute of Science, Rehovot 761000, Israel. ✉e-mail: rina.rosenzweig@weizmann.ac.il

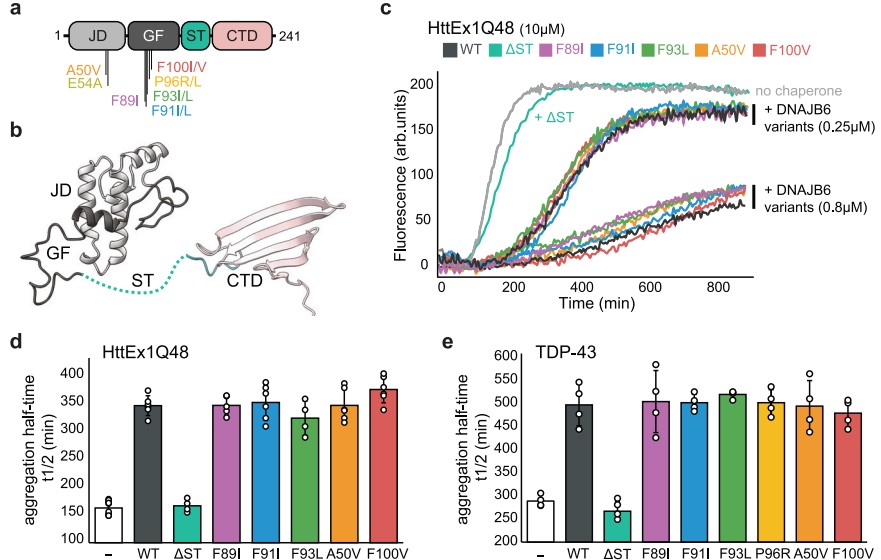

**Fig. 1 | LGMDD1 DNAJB6 mutants efficiently suppress protein aggregation.**
**a** Domain organization of DNAJB6 showing the positions of the LGMDD1-associated mutations. All mutations identified to date are located in the N-terminal section of the protein, primarily in the GF-rich region. JD is colored light gray, the GF region dark gray, the ST-rich region turquoise, and the C-terminal domain pink. **b** NMR structure of DNAJB6 lacking the ST oligomerization region (PDB 6U3R[19]), with domains colored as in (**a**). **c** Aggregation of 10 μM HTT$_{Ex1}$-Q48 alone (light gray) and in the presence of two concentrations (0.25 and 0.8 μM) of WT DNAJB6 (dark gray)

or LGMDD1 disease mutants (colored). The mutants are as efficient as the WT chaperone in preventing aggregation. **d** The effect of 0.8 μM DNAJB6 WT and disease mutants on HTT$_{Ex1}$-Q48 (10 μM) aggregation half times. Data represents mean values ± s.d (*n* = 6 independent experiments for control, DNAJB6 WT, F91L, A50V, and F100V variants; *n* = 5 for F89I; *n* = 4 for DNAJB6 ΔST and F93L variants). **e** The effect of 5 μM DNAJB6 WT and disease mutants on 10 μM TDP-43 aggregation half times. Data represents mean values ± s.d (*n* = 4 independent experiments).

Hsp70 regulation[18–20]. The C-terminal substrate-binding domain of DNAJB6 is, interestingly, not of the typical double β-barrel structure that characterizes other class B members[8,21–23], but rather a single β-sheet-rich CTD domain connected to the GF by a serine/threonine (ST) low-complexity region[19,24] (Fig. 1a, b). This ST region, which is absent from canonical JDPs, was shown to be essential for both DNAJB6 oligomerization into large complexes of tens of subunits and for the chaperone's ability to prevent aggregation[25,26].

Although the cellular functions of DNAJB6 are still under investigation, recent studies have shown that the chaperone is required for protecting the FG-Nups proteins from forming non-native structures during nuclear pore complex biogenesis[27]. DNAJB6 was also recently reported to be a potent inhibitor of aggregation and toxicity of disease-associated polyglutamine proteins in vivo[28–30]. These findings add to the growing evidence of DNAJB6 being a general protection factor, very efficiently suppressing huntingtin aggregation in the brain, as well as the formation of amyloids and amorphous aggregates in a number of cell lines[29–37]. LGMDD1 pathogenic mutations were shown to disturb this aggregation-prevention activity, evident by the accumulation of protein aggregate deposits in muscle cells[6,11], and impaired ability to suppress huntingtin polyQ and TDP-43 aggregation in model cell lines[6,13,16,30,38–41]. However, the mechanisms by which the LGMDD1-associated DNAJB6 mutations impair the chaperone's function are currently unknown.

In this study, we find that although aggregate deposits are a characteristic LGMDD1 pathology, the in vitro aggregation-prevention activity of DNAJB6 disease mutants is not affected. The pathogenic mutants, however, differ structurally from the WT protein in the J-domain/GF region. While in WT DNAJB6 the J-domain interaction with Hsp70 is regulated by the GF, we find that this regulation is disrupted in all the disease mutants identified in LGMDD1 patients. As a result, these variants can recruit and hyperactivate Hsp70 chaperones in an unregulated manner. Our findings further indicate that this high-affinity, non-productive interaction with mutant DNAJB6 depletes cellular levels of Hsp70 chaperone in *C. elegans* and disrupts protein homeostasis.

## Results

### LGMDD1 DNAJB6 mutants efficiently suppress protein aggregation

The accumulation of protein aggregates is considered to be the hallmark pathology of LGMDD1 disease, with several studies showing impaired anti-aggregation activity of DNAJB6 disease mutants in a variety of model cellular systems[6,13,16,30,31,39–41]. However, in such systems, it is usually difficult to assess if the impaired function is a direct result of diminished chaperoning activity of the disease DNAJB6 mutants, or whether this is caused by altered interactions with other downstream cellular factors. To address this, we measured the aggregation-prevention activity of different DNAJB6 LGMDD1 disease mutants in vitro.

WT DNAJB6 is among the strongest suppressors of amyloid aggregation, with the best-characterized clients of the chaperone being polyglutamine (polyQ) containing proteins and peptides[26,42–44]. Recombinant WT DNAJB6 indeed displayed very efficient aggregation-suppression activity toward huntingtin protein with 48-polyglutamine extension (HTT$_{Ex1}$-Q48), as monitored by the increase in thioflavin T (ThT) fluorescence (Supplementary Fig. 1a). Substantial delay in the onset of aggregation (lag time) was detected even at substoichiometric concentrations of the chaperone (from 0.005:1 molar ratio of DNAJB6:HTT$_{Ex1}$-Q48 and upward) (Supplementary Fig. 1a), in complete agreement with previous reports[26,31,42–44]. This anti-aggregation activity for HTT$_{Ex1}$-Q48 was not further bolstered by the addition of Hsp70 (Supplementary Fig. 1b), suggesting that DNAJB6 chaperones, in vitro, can operate in an Hsp70-independent manner.

Next, we tested the aggregation-prevention activity of DNAJB6 disease mutants toward HTT$_{Ex1}$-Q48. Six disease mutants were chosen based on their abundance in the population, disease severity and onset, variable clinical presentations, and their spatial location within the protein covering the disordered GF linker (F89I[13], F91I[13], F93I[15]), helix V (P96R[11], F100V[15]), and the J-domain (A50V[16], E54A[16]). Mutant chaperone activity toward HTT$_{Ex1}$-Q48 was measured at two DNAJB6:polyQ molar ratios (0.025:1 and 0.08:1) and compared to the

activity of the WT chaperone. DNAJB6 lacking the conserved S/T-rich region (DNAJB6 ΔST), required for aggregation prevention[42], was used as a negative control.

Surprisingly, we found that with the exception of DNAJB6 ΔST, all of the tested DNAJB6 disease mutants were as effective as WT DNAJB6 in preventing polyQ amyloid aggregation (Fig. 1c, d, and Supplementary Fig. 1c), delaying the $HTT_{Ex1}$-Q48 aggregation half-time ($t_{1/2}$; defined as the time point when the ThT intensity reaches halfway between the initial baseline and the final plateau value[45]) by 200 min, on average (Fig. 1d). Thus, disease-associated DNAJB6 mutants retain their ability to prevent the toxic polyQ aggregation, compared to the WT.

To verify that the robust chaperoning function of DNAJB6 disease mutants is not limited to aggregation-prevention of polyQ amyloid, we measured the chaperones' activity toward another documented client of DNAJB6, TDP-43. Accumulation of TDP-43 in stress granules and inclusions was detected both in LGMDD1 patients' muscle biopsies and in cell lines expressing the disease-associated DNAJB6 mutants[38,46]. In vitro, the aggregation of TDP-43 can be induced by cleaving the protein from the MBD solubility tag and monitoring the formation of aggregates via the increase in light scattering. As expected, TDP-43 aggregation was efficiently delayed by the addition of WT DNAJB6 chaperone. However, unlike in the case of polyQ amyloids, excess of the chaperone was required for efficient aggregation prevention (Supplementary Fig. 1d). Addition of similar concentrations of LGMDD1-associated DNAJB6 mutants resulted in aggregation prevention comparable to that of the WT chaperones, delaying the aggregation half-time by ~300 min (Fig. 1e).

Combined, these results show that DNAJB6 LGMDD1 mutants are not impaired in their aggregation-prevention activity and are consistent with a parallel study showing that DNAJB6 mutants also maintain their aggregation-prevention activity in cells[47].

## Regulatory region in DNAJB6 GF blocks Hsp70 binding

As DNAJB6 mutants have WT-like chaperoning activity, the accumulation of protein aggregates associated with the LGMDD1 disease must therefore stem from another functional defect in these chaperones. One such potential function is the interaction of DNAJB6 with the downstream Hsp70 chaperones, which was suggested to play a critical role in LGMDD1 pathogenesis[35].

The interaction of all members of the JDP family with Hsp70 is mediated through their conserved J-domain. This domain binds to Hsp70 and triggers ATP hydrolysis, enacting a large conformational change within the chaperone, thereby greatly increasing its affinity for substrates[7,10,48,49]. Interestingly, in DNAJB6, the GF-rich linker that connects the J-domain to the substrate-binding domains was shown to form a helical element that docks onto the J-domain and sterically blocks the potential Hsp70-binding sites[19]. A similar regulatory helical element also exists in the canonical class B JDP, DNAJB1, where this regulation is essential for DNAJB1-Hsp70 mediated disaggregation of amyloid fibers[50].

We therefore hypothesized that LGMDD1 pathogenic mutations, the majority of which are located in or near the regulatory helix, disrupt the regulation and thus impair DNAJB6-Hsp70 functions.

In order to test this, we first needed to confirm that the presence of DNAJB6 helix V in the GF indeed prevents Hsp70 from binding to the J-domain. To this end, we generated two constructs—$DNAJB6^{JD-GF}$ containing the Hsp70-binding J-domain and the inhibitory GF-region, and $DNAJB6^{JD}$ constructs, which lack the inhibitory element. Using NMR, we then evaluated the ability of these fragments to interact with Hsp70 chaperones. The addition of $^2H$ Hsp70 indeed caused significant chemical shift and intensity changes to the $^{15}N$-$^1H$ HSQC spectrum of the $^{15}N$-labeled DNAJB6 J-domain. Following a complete NMR assignment of DNAJB6 J-domain residues, we mapped the interaction of Hsp70 to its helices II and III (Fig. 2a, b). Thus, Hsp70 binds the noncanonical

class B JDPs in a conserved manner similar to that of the canonical class A and B JDPs[50].

No interaction, however, was observed between $DNAJB6^{JD-GF}$ and Hsp70, validating that the presence of helix V in the GF region indeed blocks the JD Hsp70-binding site (Fig. 2c), as suggested by Karamanos et al.[19].

## LGMDD1 mutations disrupt the JD-GF inhibition of the J-domain

Confirming that DNAJB6 GF helix V indeed blocks the Hsp70 binding sites on the J-domain and prevents Hsp70 chaperone binding, we next tested the effect of the LGMDD1 mutations on this regulatory element.

To date, 16 LGMDD1 pathogenic mutations have been reported in DNAJB6—13 are in the G/F regulatory region itself (F89I/L, F91I/L/V, F93I/L, N95I, P96R/L, D98Δ and F100I/V)[13,15], and three newly identified mutations are located in helix III of the JD (A50V and E54A/K)[16], which forms several key contacts with the GF region (Figs. 1a and 3a). Interestingly, all of the amino acids that are mutated in patients contribute to stabilizing the interaction of the inhibitory GF element with the J-domain of DNAJB6 (Fig. 3a).

To test the structural changes to the JD-GF interaction caused by LGMDD1 pathogenic mutants, we generated $DNAJB6^{JD-GF}$ constructs carrying the disease mutations. Overall, eight $DNAJB6^{JD-GF}$ variants (WT and seven mutants) were isotopically $^{15}N$-labeled and subjected to NMR measurements. The resulting $^1H$-$^{15}N$ HSQC NMR spectra of the mutants differ significantly from that of the wild-type protein, as well as from each other (Fig. 3b and Supplementary Fig. 2).

In the case of the F89I, F91I, and F93L mutants, located in the disordered GF region, large chemical shift changes were observed in helices III and IV of the J-domain and directly near the mutation site (Supplementary Fig. 2). No changes were detected in helices I and II, or in the conserved HPD motif. F89I mutant displayed the least pronounced chemical shift changes, and its overall conformation remained similar to that of the WT $DNAJB6^{JD-GF}$.

Mutations in the J-domain (E54A and A50V) or in the inhibitory helix V (P96R and F100V) caused significant chemical shift changes, covering the entire protein. Here, the most significant changes were detected in helices II, III, and V, regardless of the position of the mutation, and the chemical shifts of these proteins more resembled those of the isolated J-domain than of the WT $DNAJB6^{JD-GF}$ construct (Supplementary Fig. 2). Because chemical shifts are sensitive probes of structure, this result strongly suggests that these four disease mutants sample similar conformations as the free J-domain ($DNAJB6^{JD}$), lacking the helix V GF inhibition.

## LGMDD1 mutations destabilize the JD-GF interaction

The disruption of the JD-GF inhibition was further made evident when overlaying the WT $DNAJB6^{JD-GF}$ and mutant NMR spectra. Despite these mutations being localized at different regions of the protein, and distant in space from each other, their chemical shifts titrated in a linear fashion with the endpoint being the fully inhibited WT $DNAJB6^{JD-GF}$ and the free $DNAJB6^{JD}$ (Fig. 3b). Such behavior can be explained in terms of a simple, unifying model, whereby mutations lead to a gradual conversion of the fully inhibited WT DNAJB6 to a more uninhibited, open-like conformation, consistent with a gradual detachment of the inhibitory GF helix V. Overall, 21 residues located in helices II and III displayed such linear behavior, allowing us to calculate the fractional population of DNAJB6 in the free (uninhibited, $p_{Free}$) state, assuming a two-site exchange mechanism between inhibited, $DNAJB6^{JD-GF}$ and free, $DNAJB6^{JD}$ states that is fast on the NMR chemical shift time-scale (Fig. 3c and Supplementary Fig. 3a).

As a whole, all tested DNAJB6 LGMDD1 pathogenic mutations caused destabilization to the JD-GF interaction, and exchanged rapidly between the inhibited and free J-domain states, with their relative mixture differing between mutations. Based on the chemical shift differences between the inhibited $DNAJB6^{JD-GF}$ and the free $DNAJB6^{JD}$

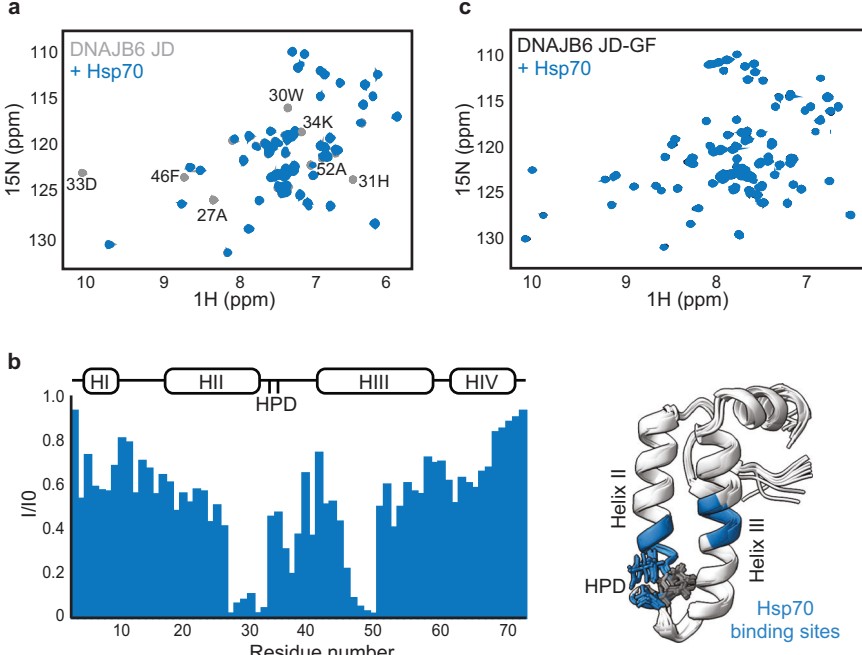

**Fig. 2 | Regulatory region in DNAJB6 GF blocks Hsp70 binding. a** $^{15}$N–$^1$H HSQC spectra of 0.2 mM DNAB6$^{JD}$ alone (light gray), and in complex with 0.1 mM $^2$H Hsp70 chaperone (blue). Peaks missing from the colored spectrum indicate proximity to Hsp70 (binding). **b** (left) Residue-resolved NMR signal intensity ratios I/I$_0$, where I and I$_0$ are signal intensities for Hsp70-bound and free DNAJB6$^{JD}$, respectively. The positions of the four helices in each J-domain are indicated at the top of the plot. Large changes in intensity are detected at the end of helix II, the flexible loop containing the conserved HPD motif, and at helix III, corresponding to Hsp70-binding sites. **b** (right) Cartoon representation of the 10 lowest energy solution-NMR structures calculated by CS-ROSETTA, with the Hsp70-binding region, identified by NMR colored blue. **c** $^{15}$N–$^1$H HSQC spectra of 0.2 mM DNAJB6$^{JD-GF}$ alone (black), and in the presence of 0.4 mM $^1$H Hsp70 (blue). No changes were observed in the spectrum of DNAJB6$^{JD-GF}$ upon the addition of Hsp70, indicating a lack of interaction.

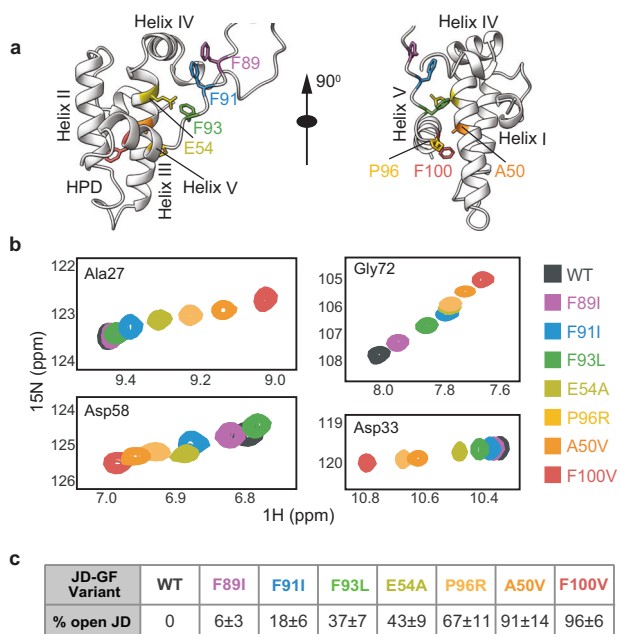

**Fig. 3 | LGMDD1 mutations disrupt the JD-GF inhibition. a** Cartoon representation of DNAJB6$^{JD-GF}$ structure (PDB 6U3R[19]) illustrating the location of the residues that are mutated in patients with LGMDD1. **b** Superposition of selected $^{15}$N–$^1$H HSQC spectral regions of wt (gray) and mutant DNAJB6$^{JD-GF}$ (colored as indicated), focusing on A28, D58, G72, and D33 residues located in Helix II, III, IV, and the conserved HPD motif, respectively. The mutants show a progressive titration of cross-peak chemical shifts toward a conformation in which the GF is undocked from the J-domain. **c** Table summarizing the fractional population of the open (uninhibited) JD-GF conformation present in each of the disease mutants. Values are means ± SEM from 21 residues.

constructs, as measured for the D33 that showed the largest CSP, we have estimated this exchange rate to be greater than 2500 s$^{-1}$.

The populations of the free J-domain varied greatly among the different LGMDD1 DNAJB6 mutants. Disease mutations located in inhibitory helix V (P96R and F100V) showed high degrees of opening, shifting the J-domain conformational equilibrium from a completely inhibited state in the WT DNAJB6$^{JD-GF}$ (p$_{Free}$ = 0) to p$_{Free}$ ~ 67% and p$_{Free}$ ~95%, respectively (Fig. 3c and Supplementary Fig. 3a). The near full population shift for the mutation in residue 100 is consistent with its location in the middle of inhibitory helix V, where it normally forms aromatic-aromatic contacts with both helix II (Y24) and III (F46) of the J-domain[19].

Mutations located in the disordered part of the GF had smaller degrees of JD-GF destabilization, with p$_{Free}$ of 18% in F91I and 37% in F93L disease mutants. Mutation to residue 89 (F89I), which only forms contacts with helix IV of the JD, led to a very minor shift from the inhibited state, with just 6% free conformation detected (Fig. 3c and Supplementary Fig. 3a). More detailed structural studies, however, are required to provide a mechanistic understanding as to how these mutations induce helix V undocking from the J-domain.

Interestingly, the A50V mutation also caused a substantial opening of the JD-GF inhibition, shifting the equilibrium almost entirely to the open state (p$_{Free}$ ~ 91%). Residue 50 is located in helix III of the JD and, based on the NMR structure[19], forms direct contacts with helix IV (Fig. 3a and Supplementary Fig. 3a). It is possible that the substitution of alanine with the bulkier valine residue generates steric interference and releases the GF inhibition. Mutation to residue 54 (E54A), which forms a salt bridge with R94 in the GF region[19] and thus stabilizes helix V docking, likewise induces opening, albeit to a lesser degree— increasing the population of the free JD to ~43% (Fig. 3c).

To further evaluate the structural changes induced onto the chaperone by the different mutations, we determined the secondary

structure for their respective variants. F89I, F91I, A50V, P96R, and F100V secondary structures were derived from the assigned backbone ($^{13}C_{\alpha}$, $^{13}C_{\beta}$, $^{13}C'$, $^{15}N$, and $^{1}H_N$) shifts using the program TALOS$^{+}$[51], and compared to that of the WT JD-GF construct[19] (Supplementary Fig. 4). Interestingly, each of the DNAJB6 constructs still formed five stable helices—four located in the J-domain (H-I to H-IV), and the fifth stable helix in the middle of the GF region (residues 96 to 104), consistent with the NMR structure of WT DNAJB6[19] (Supplementary Fig. 4). Moreover, a stable helix V was visible not only in the DNAJB6 mutants which mostly populate the inhibited conformation, but also in F100V and A50V DNAJB6, that primarily populate the uninhibited state ($p_{Free}$ ~ 95% and 91%, respectively). Helix V thus remains structured even after its detachment from the J-domain.

In summary, all LGMDD1 mutations tested destabilize the JD-GF interaction, populating new conformations in which the GF is detached from the J-domain, while helix V remains folded.

### DNAJB6 LGMDD1 mutants cause unregulated binding and activation of Hsp70

Finding that all LGMDD1 disease mutants have a conformation in which the GF is detached, suggests that, unlike the WT DNAJB6$^{JD-GF}$ proteins, these variants should be able to interact with and activate Hsp70 chaperones.

To test this, we measured the enhancement of Hsp70 ATP hydrolysis rates by WT DNAJB6 and LGMDD1 pathogenic mutants. A sensitive assay for the detection of inorganic phosphate was used to measure the weak ATPase activity of Hsp70. The assay uses a chemically modified version of the phosphate-binding protein (PBP) from *E. coli*, which undergoes a large change in fluorescence upon binding to phosphate[52,53] (Fig. 4a).

In line with previous publications, Hsp70 alone has a slow basal ATPase activity, corresponding to $0.2 \pm 0.04$ nM Pi min$^{-1}$ (Fig. 4b, white bar)[54]. The addition of WT DNAJB6$^{JD-GF}$ did not significantly affect Hsp70 ATPase activity, consistent with its J-domain being blocked by the GF region (Fig. 4b, dark gray bar). In contrast, the isolated J-domain of DNAJB6, which is constitutively open and available for Hsp70 binding, induced a dramatic rise in the rate of phosphate release, increasing the observed initial rate 8-fold to $1.74 \pm 0.18$ nM Pi min$^{-1}$ (Fig. 4b, light gray bar).

The degree of Hsp70 activation by the disease mutants correlated with their respective population of the free (uninhibited) J-domain conformation (Supplementary Fig. 3b). The largest activation was measured for A50V ($p_{Free}$ ~ 91%) and F100V ($p_{Free}$ ~ 95%) mutants, which increased the rate of Hsp70 Pi release by roughly 6-8-fold, with an observed initial rate of $1.24 \pm 0.15$ nM and $1.64 \pm 0.19$ nM Pi min$^{-1}$, respectively (Fig. 4b). The F89I, F91I, and F93L mutants, located in the disordered region of the GF, showed only a moderate degree of Hsp70 activation, increasing the Pi release rates by 2- to 3-fold. While, addition of DNAJB6$^{JD-GF}$ E54A and P96R mutants ($p_{Free}$ of 53% and 67% respectively), resulted in a roughly 4-fold enhancement of Hsp70 ATP hydrolysis rate.

In summary, all LGMDD1 disease mutants show loss of the JD-GF regulation and can therefore enhance Hsp70 ATP hydrolysis rates, albeit to a different degree.

To determine if this difference in the ability of the different pathogenic mutants to activate Hsp70 stems from the relative population of the free J-domain state, and not from differences in their binding modes to the chaperone, we monitored DNAJB6$^{JD-GF}$ interaction with Hsp70 by NMR (Fig. 4c–j).

$^{1}H$-$^{15}N$ HSQC experiments were recorded for the DNAJB6$^{JD-GF}$ construct of each of the seven DNAJB6 disease mutants alone, and in the presence of Hsp70 chaperones. In order to ensure that Hsp70 remains in the ATP-bound state, which has a high affinity for the JDPs, a variant bearing a T204A mutation, which significantly slows down basal Hsp70

ATP hydrolysis rates, was used, along with an ATP regeneration system[55,56].

No changes were detected to the resonances of the WT DNAJB6$^{JD-GF}$, even upon addition of 4-fold excess of fully protonated Hsp70, consistent with the J-domain in this construct being inaccessible for Hsp70 binding (Fig. 4c). In contrast, all of DNAJB6$^{JD-GF}$ constructs carrying LGMDD1 mutations displayed noticeable intensity changes in residues located in helices II, III, and the conserved HPD motif, upon addition of 0.25–1.0 molar ratio of Hsp70 (Fig. 4d–j). As expected, there was an inverse correlation between the amount of Hsp70 that was required to detect significant peak broadening to the NMR spectrum and the fraction of the free JD-GF conformation for each DNAJB6$^{JD-GF}$ variant (1:1 DNAJB6$^{JD-GF}$:Hsp70 ratio for F89I vs 1:0.25 ratio for A50V and F100V mutants). Overall, the observed mode of Hsp70 binding to these mutants was similar to the interaction with J-domains in other JDPs, suggesting that the disease mutations do not affect the binding interface, but rather cause partial or full loss of the JD-GF inhibition, freeing the J-domain for Hsp70 binding.

Interestingly, besides the conserved binding site, the addition of Hsp70 to the mutant DNAJB6$^{JD-GF}$ constructs also caused changes to the chemical shifts of residues 70-74 located in the GF region (Fig. 4d–j). To rule out that these changes are due to a direct interaction between the GF region and Hsp70, we performed NMR binding experiments between DNAJB6$^{JD-GF}$ F93L and A50V mutants and $^{13}CH_3$-ILVM labeled Hsp70. No additional binding regions were detected in Hsp70 upon the addition of DNAJB6$^{JD-GF}$ F93L and A50V mutants compared to the binding of DNAJB6JD (Supplementary Fig. 5a–c). Furthermore, no binding was detected between $^{15}N$-labeled DNAJB6$^{JD-GF}$ F93L and A50V mutants and Hsp70 in the ADP-bound state, indicating that the GF region does not bind Hsp70 as a client (Supplementary Fig. 5d, e).

The observed decrease in the GF residue intensities thus most likely arises from an allosteric effect caused by conformational changes due to the detachment of the GF from the J-domain in the process of Hsp70 binding. Reduction in signal intensity was also observed in residues within the GF region, particularly for mutants that predominantly populate the GF-inhibited conformation (Fig. 4d–g). These changes likely reflect alterations in the overall conformation of this region due to the detachment of helix V.

In summary, while the WT DNAJB6$^{JD-GF}$ is found in an inhibited conformation and does not bind and activate the Hsp70 chaperone, all LGMDD1 mutants lack such inhibition, leading to unregulated enhancement of Hsp70 ATP hydrolysis rates.

### The JD-GF inhibition, preventing Hsp70 binding, is maintained in the full-length DNAJB6

After identifying the disruption of JD-GF inhibition is common to all the LGMDD1 mutant JD-GF constructs, we next expanded our testing to the full-length DNAJB6 variants.

Structural characterization of full-length DNAJB6 chaperones by NMR, however, is challenging, as they form large ~1MDa oligomeric assemblies (Supplementary Fig. 6a). We therefore generated a truncated monomeric version of DNAJB6 by removal of the ST region, following a protocol from Karamanos et al.[19], allowing us to obtain high-quality $^{1}H$-$^{13}C$ HMQC spectra of $^{13}CH_3$-ILVM labeled DNAJB6 monomer (DNAJB6 ΔST, referred from this point as DNAJB6$^{mono}$) (Supplementary Fig. 6b, c). Comparison of the J-domain methyl peaks in DNAJB6$^{mono}$ to those of the isolated DNAJB6$^{JD}$ or DNAJB6$^{JD-GF}$ construct showed that the full-length protein is much more similar to the JD-GF than to the free J-domain (Supplementary Fig. 6d–g). Thus, in agreement with previous reports[19], in the presence of other DNAJB6 domains, the J-domain is still blocked by helix V of the GF region. Addition of Hsp70 to DNAJB6$^{mono}$ did not cause any measurable changes to chemical shifts or intensity of the proteins, indicating that

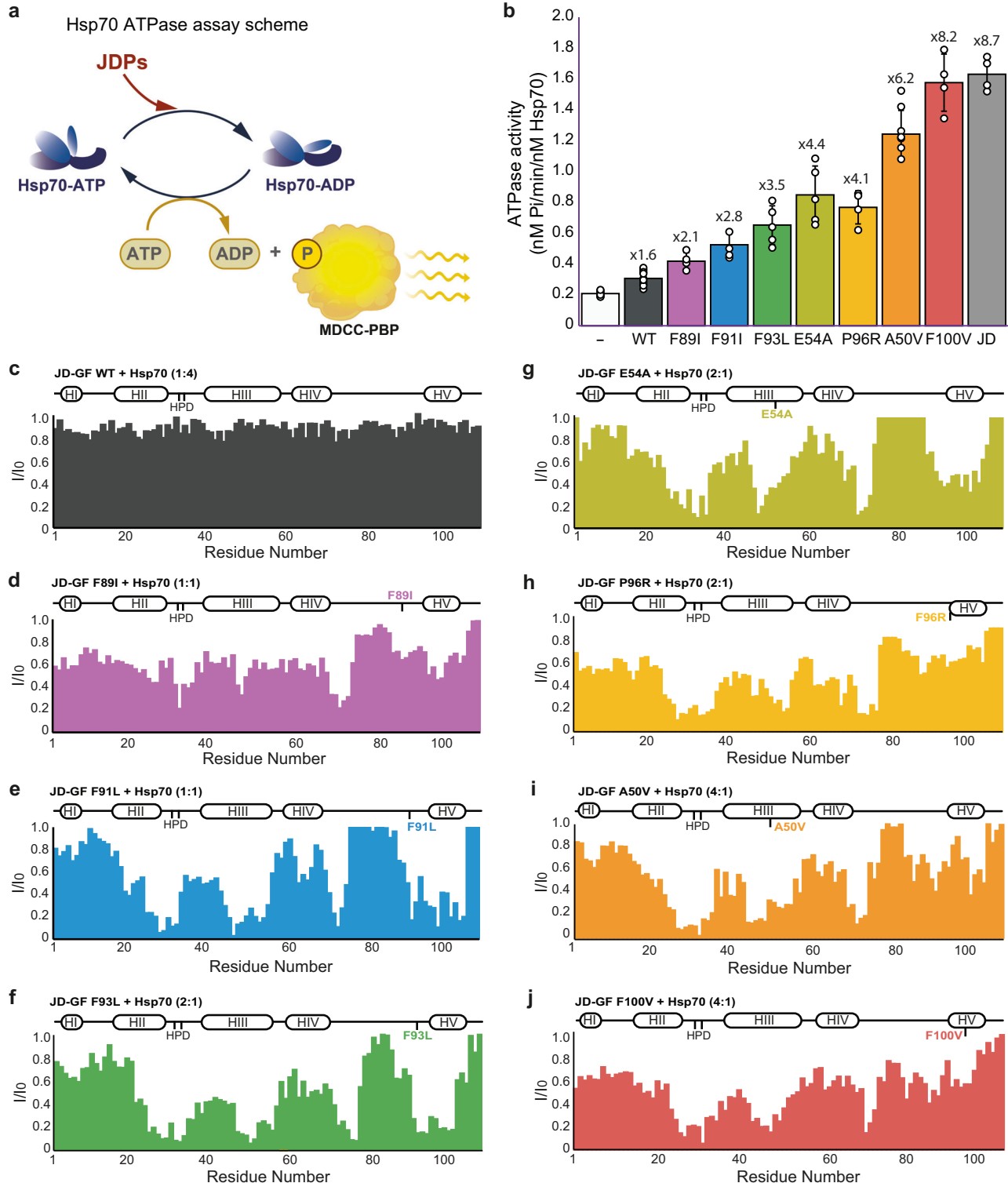

**Fig. 4 | DNAJB6 LGMDD1 mutants cause unregulated binding and activation of Hsp70. a** Schematic representation of the Hsp70 chaperone cycle highlighting the JDP-induced ATP hydrolysis and the measurement of ATP hydrolysis rates using the fluorescently labeled phosphate-binding protein (PBP). **b** Steady-state ATPase activity of Hsp70 alone (white), and Hsp70 incubated with WT DNAJB6$^{JD-GF}$ (dark gray) or LGMDD1-associated mutants (colored). All the disease mutants enhanced Hsp70 ATP hydrolysis rates 2-8-fold, while only a minor, 1.6-fold activation was detected upon the addition of WT DNAJB6$^{JD-GF}$ to Hsp70. Data are means ± SEM (n = 8 independent experiments for Hsp70 alone and DNAJB6$^{JD-GF}$ WT; n = 6 for DNAJB6$^{JD-GF}$ F93I, E54A, and A50V; n = 5 for F89I, F91I and P96R; n = 4

for DNAJB6$^{JD-GF}$ F100V and DNAJB6$^{JD}$). **c–j** Residue-resolved NMR signal intensity ratios $I/I_0$, where $I$ and $I_0$ are signal intensities for Hsp70-bound and free WT DNAJB6$^{JD-GF}$ (**c**), F89I (**d**), F91I (**e**), F93L (**f**), E54A (**g**), P96R (**h**), A50V (**i**), and F100C (**j**) variants, respectively. The positions of the four helices in each J-domain are indicated at the top of the plot. Large changes in intensity are detected at the end of helix II, the flexible loop containing the conserved HPD motif, helix III, and residues 70-74 of the GF, corresponding to Hsp70-binding sites. With the exception of WT DNAJB6$^{JD-GF}$ which does not interact with Hsp70, Hsp70-binding in all the disease mutants involved the same JD regions as WT DNAJB6$^{JD}$ (Fig. 2b), indicating a conserved mode of interaction with the chaperone.

just like in the case of the JD-GF construct, full-length DNAJB6 is found in an inhibited state and does not bind to Hsp70 (Fig. 5a, c).

Next, we tested the effect of LGMDD1 mutations on the structure of DNAJB6$^{mono}$ and its interaction with Hsp70. The disease A50V mutation was chosen, as it caused almost a complete release of the GF from the J-domain in the DNAJB6$^{JD-GF}$ construct, while minimally affecting the GF structure itself (Fig. 3). The methyl spectrum of DNAJB6$^{mono}$ A50V was indeed substantially different from that of the WT protein, with large chemical shifts being detected in the J-domain and GF region of the protein (Supplementary Fig. 7a, b). No changes were detected in other regions of the protein, strengthening our observation that the CTD client-binding region of DNAJB6 is not affected by the pathogenic mutations (Supplementary Fig. 7a, b). The overall conformation of the DNAJB6$^{mono}$ A50V JD-GF region resembled that of the isolated J-domain (Supplementary Fig. 7c, d), demonstrating that this mutation also leads to a complete release of the inhibition in the full-length DNAJB6 protein.

In contrast to the WT protein, addition of Hsp70 to the uninhibited A50V DNAJB6$^{mono}$ caused significant decreases in NMR peak intensities, indicating a strong interaction of this LGMDD1 mutant with Hsp70 (Fig. 5b). The interaction was localized to the J-domain region, with no binding detected between Hsp70 and the DNAJB6 C-terminal substrate-binding domain (Fig. 5d). Thus, the interaction of Hsp70 with DNAJB6, a noncanonical class B JDP family member, is vastly different from that of the canonical class B DNAJB1, which binds the chaperone via an additional site located in its CTDI domain[50,57–59].

Consistent with DNAJB6$^{mono}$ A50V mutant having a constitutively uninhibited J-domain and interacting strongly with Hsp70, the mutant also caused a 7-fold activation of Hsp70 ATPase activity, while WT DNAJB6$^{mono}$ showed no such activation, as expected from its J-domain being blocked by the GF region (Fig. 5e).

We next set out to repeat the Hsp70 activation assays with full-length, oligomeric DNAJB6 chaperones. Both WT and LGMDD1 mutant chaperones were stable and assembled into large, 1MDa oligomers, as evident from the SEC-MALS measurements (Supplementary Fig. 8a, b).

In line with our observations for the monomeric construct, the full-length oligomeric WT DNAJB6 was also unable to increase Hsp70 ATP hydrolysis rates, suggesting that the JD-GF inhibition is maintained upon DNAJB6 oligomerization (Fig. 6a, gray). The very moderate 1.4 activation detected for the WT was similar to that observed when using DNAJB6 J-domain HPD mutant (DNAJB6$^{QPN}$), which cannot bind to Hsp70, and thus likely represents residual non-specific ATPase activity (Fig. 6a, cyan).

The addition of oligomeric LGMDD1 DNAJB6 mutants to Hsp70, however, resulted in various degrees of activation, ranging from 2- to 8-fold (Fig. 6a), with the most significant enhancements to Hsp70 hydrolysis rates being recorded for the A50V and F100V disease mutants, in which the J-domain is primarily found in the free conformation (Supplementary Fig. 8c). Interestingly, the F89I and F91I

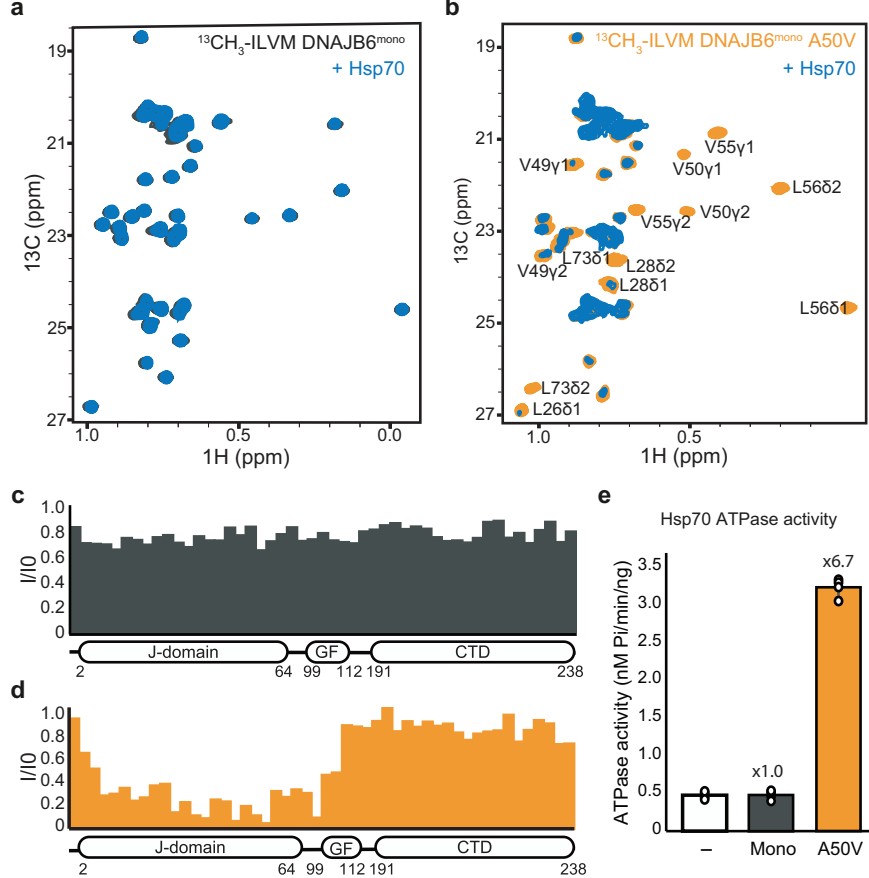

**Fig. 5 | Interaction of monomeric DNAJB6 variants with Hsp70. a** $^{13}$C-$^1$H HSQC spectra of monomeric DNAJB6 (DNAJB6 ΔST) alone (gray), and in the presence of 2-fold excess of Hsp70 (blue). No binding is detected between Hsp70 and WT DNAJB6. **b** $^{13}$C-$^1$H HSQC spectra of monomeric DNAJB6 A50V mutant alone (orange), and in the presence of a 2-fold excess of Hsp70 (blue). Significant changes to the DNAJB6 A50V spectrum were observed upon the addition of Hsp70, indicating binding. **c, d** Methyl-group peak intensity ratios ($I_{bound}/I_{free}$) of monomeric WT DNAJB6 (c) or A50V mutant (d), upon addition of Hsp70. Hsp70 caused considerable reductions in the intensity of the J-domain residues of the DNAJB6 A50V mutant, while no binding was detected with the WT protein. **e** Steady-state ATPase activity of Hsp70 alone (white) and upon incubation with monomeric WT DNAJB6 or A50V disease mutant. The WT shows no activation of Hsp70 ATPase activity, while the A50V mutant enhances the activity 6.7-fold. Data are means ± SEM ($n = 3$ independent experiments).

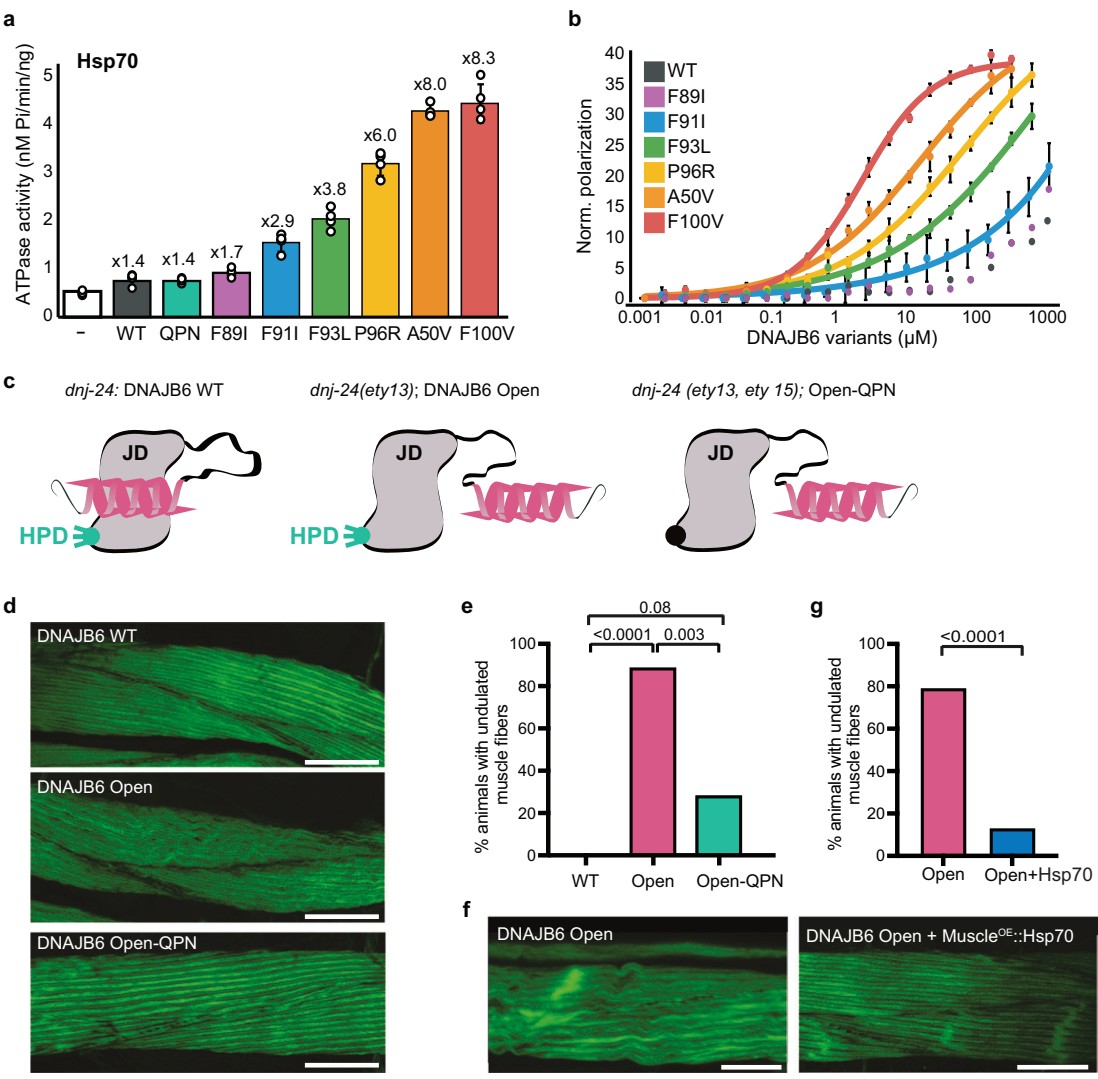

**Fig. 6 | DNAJB6 mutants display toxic gain of function through unregulated interaction with Hsp70 chaperones.** Steady-state ATPase activity of Hsp70 alone (white), and incubated with DNAJB6 WT (dark-gray) or LGMDD1-associated mutants (colored). DNAJB6$^{QPN}$ mutant ($^{31}$HPD$^{33}$ > $^{31}$QPN$^{33}$), deficient in Hsp70 binding, was used as a negative control (cyan). Data are means ± SEM ($n$ = 3). **b** Titration of Hsp70$^{AF488}$ with full-length DNAJB6 WT (dark-gray) or LGMDD1-associated disease mutants (colored), monitored by steady-state fluorescence anisotropy shown in arbitrary units. Data are presented as mean values ± SEM, ($n$ = 3 independent experiments). **c** Schematic representation of the different *C. elegans* DNAJB6 variants used. DNAJB6 WT corresponds to DNJ-24 protein with intact JD-GF inhibition and Hsp70 binding sites. DNAJB6$^{Open}$ variant corresponds to DNAJ-24 in which 4 residues in the GF helix V were mutated to release the JD-GF inhibition (see Supplementary Fig. 9 for details). DNAJB6$^{Open-QPN}$ is a double DNAJ-24 mutant that contains the mutations artificially lifting the JD-GF inhibition and mutations that abolish the ability of this variant to bind to Hsp70 (DNJ-24$^{QPN}$). **d** *C. elegans* muscle-

fiber morphology visualized by Phalloidin-FITC staining, labeling F-actin for DNAJB6 WT, DNAJB6$^{Open}$ mutant, and DNAJB6$^{Open-QPN}$ double mutant. Animals expressing WT DNAJB6 display healthy muscle morphology, while DNAJB6$^{Open}$ alters muscle morphology to disorganized myofibrils with an undulated appearance. The introduction of a second mutation to the Hsp70 binding region, DNAJB6$^{Open-QPN}$, rescues the abnormal phenotype, restoring healthy muscle morphology. Scale bars represent 10 μm. **e** Quantification of animals with the undulated phenotype from (**d**). Number of animals: Wild-type, $n$ = 15; DNAJB6$^{Open}$, $n$ = 18, DNAJB6$^{Open-QPN}$, $n$ = 16. Statistics is Fisher's exact test followed by Dunn's multiple comparison test. **f** *C. elegans* muscle-fiber morphology visualized by Phalloidin-FITC staining, labeling F-actin for DNAJB6$^{Open}$ *(dnj-24(ety13))* and DNAJB6$^{Open}$ with muscle-specific overexpression of Hsp70 *(myo-3::hsp-1)*. Overexpression of Hsp70 restores the healthy phenotype of the muscles. **g** Quantification of animals with the undulated phenotype from (**f**). Statistics is Fisher's exact test, two-tailed. Number of animals: Wild-type, $n$ = 23; DNAJB6$^{Open}$, $n$ = 18, DNAJB6$^{Open-QPN}$, *myo-3::hsp-1* $n$ = 24.

mutations, which have a relatively small percentage of free J-domain, were still able to bind to Hsp70 and increase its activity 2- to 3-fold (Fig. 6a and Supplementary Fig. 8c). Overall, all LGMDD1 DNAJB6 mutants bound and activated Hsp70 chaperones in an unregulated manner, both as monomers, and in the context of full-length oligomeric DNAJB6 chaperone constructs.

**DNAJB6 disease mutants display unregulated, high-affinity binding to Hsp70**

To learn more about the interaction of the oligomeric DNAJB6 disease mutants with Hsp70, we measured the affinities between the full-

length chaperones using fluorescence polarization anisotropy (Fig. 6b). Addition of even 10,000-fold excess of WT DNAJB6 to Alexa-488 labeled Hsp70 (1 mM vs 100 nM concentrations, respectively) caused only a very small increase in the fluorescence polarization, consistent with binding at very low affinity ($k_D$ > 10 mM). The affinity of the F89I mutant ($p_{Free}$ = ~6%) to Hsp70 was also very low, and not measurable in our experimental setting. Interestingly, this mutant was still able to activate Hsp70 chaperones (Fig. 6a), albeit to a small degree, suggesting that even very transient interactions are sufficient to trigger the conformational changes in Hsp70 necessary to activate the ATPase.

The affinity of F91I ($p_{Free}$ = 25%) and P93L ($p_{Free}$ = 32%) to Hsp70 was also very low ($k_D$ = 600 μM and 290 μM, respectively; Fig. 6b). Higher affinities of 47 μM and 9 μM were recorded for P96R ($p_{Free}$ = 67%) and A50V ($p_{Free}$ = 91%), with this binding being comparable in strength to the interaction recorded between Hsp70 and other JDPs, such as DNAJA2 and DNAJB1[50,60]. The greatest affinity ($k_D$ = 1.5 μM) was measured for the F100V mutant ($p_{Free}$ = 96%; Fig. 6b) and was over an order of magnitude tighter than a "typical" JDP-Hsp70 interaction. This may result from the avidity effect associated with the oligomeric nature of DNAJB6, which simultaneously presents multiple free J-domains for binding.

Overall we find that unlike the WT DNAJB6, which in the absence of other factors does not interact with Hsp70, all LGMDD1 disease mutants display unregulated binding to Hsp70 chaperones (Fig. 6b), highly correlated to the degree of JD-GF opening (Supplementary Fig. 8d). We therefore conclude that it is not the loss of DNAJB6 function that leads to LGMDD1 disease, but rather this unregulated interaction of DNAJB6 pathogenic mutants with Hsp70 chaperones.

### DNAJB6 mutants display toxic gain of function through unregulated interaction with Hsp70 chaperones

To investigate whether the disruption of the JD-GF inhibition is indeed responsible for the defects in muscle integrity observed in LGMDD1 patients, we used CRISPR-Cas9 to endogenously introduce mutations that artificially alleviate this JD-GF inhibition into the *C. elegans* DNAJB6 homolog, *dnj-24* (Supplementary Fig. 9a–d). *C. elegans* is a widely used multicellular model organism for studying tissue-specific chaperone networks and proteostasis[61,62]. Furthermore, while there are four noncanonical human class B JDPs, DNJ-24 represents the only member of this family found in nematodes. DNJ-24 is expressed in various tissues, with particularly high expression in muscle cells[63], and shares significant homology with DNAJB6, especially in the regions of LGMDD1 mutations (aside from F91I, Supplementary Fig. 9a–c). DNJ-24 is therefore an ideal candidate for examining the impact of DNAJB6 mutations on muscle integrity in an organismal setting.

An artificially open DNJ-24 mutant (DNAJB6$^{Open}$), lacking JD-GF inhibition, was designed by substituting four conserved residues in GF helix V with glycines or serines (Fig. 6c and Supplementary Fig. 9d) in DNJ-24 using CRISPR-Cas9. These mutations were specifically designed so they did not affect the residues found to be mutated in LGMDD1 patients, and thus any effects could be attributed solely to the disruption of the interaction between GF helix V and the J-domain.

When recombinantly expressed, this DNAJB6$^{Open}$ mutant was just as efficient as WT DNAJB6 in preventing TDP-43 aggregation in vitro (Supplementary Fig. 10a). However, unlike the WT, which showed no enhancement of Hsp70 ATP hydrolysis rates, DNAJB6$^{Open}$ hyperactivated the chaperone 5-fold (Supplementary Fig. 10b). NMR analysis of the monomeric version of DNAJB6$^{Open}$ showed that it has a completely free J-domain, with no observable docking of the GF onto the JD (Supplementary Fig. 10c–e). This result is consistent with the DNAJB6$^{Open}$ J-domain lacking all GF inhibition, allowing unregulated Hsp70 binding and activation.

Seeing that the chaperoning activity of the DNAJB6$^{Open}$ is maintained in vitro, we next tested the effect of this mutant on *C. elegans* muscle morphology in vivo. To this end the muscle fibers of worms endogenously expressing either wild-type (*dnj-24*) or DNAJB6$^{Open}$ mutant (*dnj-24(ety13)*) were stained with Phalloidin and analyzed via confocal microscopy. Interestingly, expression of the DNAJB6$^{Open}$ mutant in muscles resulted in a striking phenotype, characterized by highly disorganized muscles with undulated myofibrils, compared to the straight and parallel muscle fibers observed in wild-type worms (Fig. 6d, e). This DNAJB6$^{Open}$ phenotype is reminiscent of patient biopsies and other model organisms expressing disease-associated DNAJB6 mutants[13,35,64–66]. These findings thus confirm that the LGMDD1 disease phenotype in muscles is indeed caused by loss of GF inhibition.

It is important to note that no gross locomotion defects were detected in mutant worms in a thrashing assay (Supplementary Fig. 9e), indicating that unlike in human patients, this type of structural defect in muscle integrity is not sufficient to disrupt mobility in *C. elegans*.

We next investigated whether it is the unregulated interaction between the uninhibited DNAJB6 mutants and Hsp70 that leads to the disease phenotype. To test this hypothesis, we introduced mutations to the conserved HPD motif in the J-domain of DNJ-24 (DNAJB6$^{QPN}$), which were previously shown to abolish the JD-Hsp70 interaction (Fig. 6c and Supplementary Fig. 9d)[35,67,68]. If unregulated recruitment of Hsp70 by the DNAJB6$^{Open}$ mutant is indeed the cause of defects in muscle morphology, these additional mutations, that disrupt the DNAJB6-Hsp70 interaction, should suppress this phenomenon. Excitingly, expression of the double DNAJB6$^{Open-QPN}$ mutant in worms led to the restoration of the wild-type muscle phenotype, with the undulated muscle morphology significantly diminished (Fig. 6d, e). This result is consistent with previous studies demonstrating that inhibitors blocking JDP-Hsp70 binding can partially restore muscle performance in mouse models expressing mutant DNAJB6[35], and provides conclusive evidence that the unregulated interaction between mutant DNAJB6, lacking the GF inhibition, and Hsp70 is the cause of the disease phenotype.

However, the mechanism by which the unregulated interaction of pathogenic mutant DNAJB6 with Hsp70 causes the disease phenotype remained unclear. One possibility is that the regulation of Hsp70 binding is designed to recruit the chaperone only to specific clients in muscle cells. Disruption of such regulation would therefore result in the recruitment of Hsp70 chaperones to clients that should not be remodeled, disrupting protein homeostasis and inducing protein aggregation. Alternatively, the high affinity and non-productive interaction with DNAJB6 disease mutants could effectively trap Hsp70 chaperones, leading to broader disruption of Hsp70-dependent processes and the accumulation of protein aggregates within the muscle fibers.

To distinguish between these possibilities, we examined the effect of Hsp70 overexpression on muscle morphology. If the disease phenotype is caused by Hsp70 depletion, Hsp70 overexpression should alleviate the phenotype. On the other hand, if Hsp70 chaperones are recruited to the wrong clients, overexpression of Hsp70 should have no effect, or even exacerbate the phenotype. We selectively overexpressed the heat-inducible *C. elegans* Hsp70, *hsp-1* in muscle cells using the muscle-specific promoter *myo-3* in the background of endogenously expressed WT *dnj-24*/DNAJB6 or *dnj-24*/DNAJB6$^{Open}$ mutant. Nematodes with WT DNAJB6 had a healthy muscle appearance, and overexpression of Hsp70 in the background of WT DNAJB6 had no effect on this muscle morphology (Fig. 6f and Supplementary Fig. 9f, g). In contrast, cells expressing DNAJB6$^{Open}$ formed highly disorganized muscles with undulated myofibrils, characteristic of LGMDD1 (Fig. 6f, g). Overexpression of Hsp70, though, substantially reduced the appearance of the disorganized fibers and restored the healthy straight muscle appearance (Fig. 6f, g, Supplementary Fig. 9f).

Thus, LGMDD1 disease results from the depletion of Hsp70 chaperone levels caused by the unregulated interaction of this chaperone with the DNAJB6 mutant, which loses its JD-GF inhibition. This leads to the disruption of Hsp70-dependent processes and a subsequent decline in proteostasis.

## Discussion

Mutations in the gene encoding for DNAJB6 chaperone cause dominantly inherited muscle disease LGMD type 1D. To date, 16 pathogenic mutations have been identified in DNAJB6, all resulting in similar changes to myocytes, characterized by severe myofibrillar disintegration and accumulation of protein aggregates[11–14]. Surprisingly, we find that despite protein aggregation being the prominent phenotype of LGMDD1, the disease-causing DNAJB6 mutations do not impair the

aggregation-suppression activity of the chaperone in vitro. In fact, all seven disease mutants we tested exhibited chaperoning activity levels equivalent to the WT both in preventing the amyloid aggregation of polyQ-enriched httEx1 proteins and suppressing the liquid-to-solid transition of TDP-43. These results are consistent with a parallel study showing that DNAJB6 mutants likewise do not display major defects in reducing polyQ aggregation in cells[47] unless the substrate-binding regions are impaired[47]. This also aligns with the fact that all of the identified LGMDD1 mutations are located at the N-terminal J-domain and GF regions of DNAJB6, which are not required for client interaction[31,36,42].

Instead, we find that the LGMDD1 pathogenic mutations cause structural changes to the JD-GF regions of the chaperone. In WT DNAJB6, the GF region forms an inhibitory helix that docks onto the J-domain, preventing the chaperone from binding and activating the Hsp70 machinery. All LGMDD1 mutations, however, disrupt this autoinhibitory JD-GF mechanism, leading to unregulated binding and hyperactivation of Hsp70 in the absence of clients.

Our findings indicate that this high-affinity, non-productive interaction with mutant DNAJB6 depletes cellular levels of the Hsp70 chaperone, disrupting protein homeostasis and leading to protein aggregation. Therefore, it is not, as previously suggested, the loss-of-function of DNAJB6 that underlies the disease, but rather the unregulated binding of DNAJB6 LGMDD1 mutants to Hsp70. This hypothesis is further supported by recent findings demonstrating that inhibitors of the JD-Hsp70 interaction can reduce the severity of LGMDD1 disease[35], and explains previous observations that over-expression of WT DNAJB6 has no corrective effect on muscle morphology[13,35].

Interestingly, a similar impairment in cellular function due to a loss of JD-GF inhibition was also detected for another class B JDP family member, DNAJB1. Disruption of the inhibition and freeing of the JD abolished the ability of DNAJB1 to simultaneously recruit multiple Hsp70s to amyloid fibers, diminishing the disaggregation activity of this chaperone system[50,69].

In DNAJB1, the JD-GF inhibition is regulated through a second Hsp70-binding site, located in the CTDI domain of the chaperone[50]. Such a site, however, is absent in DNAJB6, raising the question of how the JD-GF inhibition is indeed released and regulated in the WT DNAJB6 protein. Karamanos et al.[19] suggested that the release could perhaps be mediated by the interaction of client proteins with the client-binding domain of DNAJB6. The DNAJB6 chaperone is thought to contain two client-binding domains with distinct client specificities: the amyloid binding ST-rich region[25,31,42,44,70], and the poorly characterized CTD, reported to interact with misfolded proteins prone to amorphous aggregation[36]. Therefore, it may be that client binding to only one of these domains can release the inhibition. In such a case, interaction with some clients would release the inhibition and transfer the proteins to Hsp70 chaperones, while that of others, that bind to the second domain, would not. For those clients that do not release the inhibition, DNAJB6 chaperones may instead function in an Hsp70-independent manner. The dependence of DNAJB6 on the Hsp70 chaperone machinery in the cell could thus potentially be predetermined by the type of client.

When LGMDD1 disease mutants disrupt the regulation in DNAJB6, though, this allows the chaperone to constantly recruit Hsp70 chaperones even when not bound to clients. This interaction, which we find occurs with highly increased affinity compared to the canonical JDP-Hsp70 binding[17,50,71], then depletes the overall levels of Hsp70 chaperone in the muscle, perturbing protein homeostasis. Hsp70 is necessary for the maintenance and regeneration of muscle fibers, ensuring proper folding of myofibrillar proteins, particularly at the Z-discs, which experience significant stress and strain during muscle contractions[72]. Depletion of Hsp70 chaperone levels would therefore result in inefficient protein quality control in the muscles and an increased accumulation of misfolded myofibrillar proteins, such as desmin[64,73,74]. This could then explain the muscle weakness and widespread protein aggregation detected in LGMDD1 patients. We, however, did not detect observable locomotion defects in mutant C. elegans animals, suggesting that in this model organism, the structural defects in muscle integrity are not sufficient to disrupt mobility. This is most likely due to the differences in muscle structure and physiology between worms and humans[75]. Testing the effect of LGMDD1 mutations in additional model organisms is therefore needed to shed light on the complex pathological implications of these mutations.

One of the puzzling aspects of LGMDD1 is that, despite DNAJB6 being a ubiquitously expressed protein, its pathogenic mutations selectively impact skeletal muscle, while the central nervous system remains largely unaffected. Our findings linking the disease phenotype to the depletion of cellular Hsp70 chaperone levels caused by its unregulated, high-affinity interaction with DNAJB6 LGMDD1 mutants, may shed light on this issue. The high expression of DNAJB6 in muscles, compared to the brain and nervous system, combined with the decreased turnover rate and elevated levels of mutant DNAJB6[13,35,64,76,77] (Supplementary Fig. 9f), could explain how these mutants deplete Hsp70 levels selectively in the muscle tissue. Conversely, Hsp70 expression is higher in the central nervous system compared to skeletal muscle[77,78], potentially providing a buffer or compensatory mechanism to partially offset the effects of DNAJB6 mutations, thereby maintaining protein homeostasis and preventing the manifestation of the disease phenotype.

Notably, LGMDD1 is not the only chaperonopathy involving Hsp70 dysregulation specifically affecting skeletal muscles. A recent study exploring the pathogenic mechanism of mutations in BAG3, associated with myofibrillar myopathy (MFM), showed striking similarities. There, mutations in BAG3 were shown to inhibit Hsp70 client processing and deplete Hsp70 chaperone levels via aggregation[79]. Interestingly, blocking the high-affinity Hsp70-BAG3 interaction was shown to partially rescue phenotypes[79,80], suggesting that, like LGMDD1, BAG3-associated myopathy is also due to a dominant interaction with Hsp70. The similarities between these findings suggest that more myopathies may share a common molecular feature of aberrant Hsp70-dependent protein quality control. Impeding these abnormal chaperone interactions may therefore potentially present a general approach to restoring muscle function in myopathies.

While broad inhibition of JDP-Hsp70 interactions was shown to restore muscle strength in mice[35], such inhibitors would also be widely detrimental to the many Hsp70 cellular functions, possibly leading to severe side effects. In contrast, selectively targeting the newly found regulation mechanism in DNAJB6 or the DNAJB6 JD-Hsp70 interaction, could prove a significantly safer approach, enabling the development of new and effective treatments for LGMDD1.

## Methods

### Construct preparation

Codon-optimized DNAJB6$^{WT}$ (isoform b), DNAJB6$^{JD}$ (res. 1-72), DNAJB6$^{JD-GF}$ (res. 1-109), DNAJB6$^{\Delta ST}$ (deletion of res. 138-182), and their derivatives were expressed from the pET-SUMO vector with an N-terminal His$_6$ purification tag. TDP-43 with a N-terminal MBP-HIS$_6$ tag and a TEV cleavage site, was a gift from Nicolas Fawzi (Addgene plasmid #104480). HTTEx1-Q48 with an N-terminal GST fusion was a gift from Janine Kirstein. pET22b_PstS_1 plasmid encoding for the *E. coli* phosphate-binding protein (PBP) with A197C mutation was a gift from Martin Webb (Addgene plasmid #78198).

All mutations were introduced by QuickChange or TPCR.

### *C. elegans* strains

Wild-type strains were *C. elegans* variety Bristol, strain N2. Worms were maintained according to standard methods[81]. Worms were grown at 20 °C on nematode growth media (NGM) plates seeded with bacteria

(*E. coli* OP50) as a food source. Strains generated in this study: MOS639 *dnj-24(ety13)III; him-5(e1490)V* (2Xbackcrossed), MOS664 *dnj-24(ety13-ety15)III* (2Xbackcrossed); *him-5(e1490)*, MOS760 *etyEx296[myo-3::hsp-1 40 ng/μL, myo-2::mCherry 5 ng/μL, pBS 50 ng/μL]*, MOS761 *dnj-24(ety13); etyEx296*.

## CRISPR modifications in *C. elegans*

CRISPR strategy was based on Paix et al.[82]. Recombinant SpCas9 (IDT) was injected into N2 gonads with tracrRNA (IDT) and two crRNAs, one targeting the *dpy-10* locus[82] and one targeting the *dnj-24* locus (ACACGTTCTCAAATGGATCC). Repair template was the ΔH5 (DNJ-24$^{Open}$) ssODN CACAGTCACGATATGTTCCGATCACCTTTTGATGGTGG TCGCGAGGGCTCC GGCAACAGGGATCCATTTGAGAACGTGTTTTTCG ACGACGCGTTCAC. Underlined nucleotides are mutations introduced to change the helix V region (from **IFREFF to GG**RE**GS**) and cancel the PAM sequence, and are surrounded by two 35nt-long homology arms. Rol/dpy F1 progeny were screened by PCR to identify the *dnj-24(ety13)* ΔH5 allele. Each animal was subjected to two diagnostic reactions: GTGGAAATGGCATGAGGACT + GCCGGAGCCCTCGCGACCACC, specific for the CRISPR *ety13* allele, and GTGGAAATGGCATGAGGACT + GCCGAAGAACTCGCGAAAAAT, specific for the WT allele. Animals positive for the ΔH5 edit were homozygozed and backcrossed to dispose of the *dpy-10* allele.

To add the $^{31}$HPD$^{33}$ > $^{31}$QPN$^{33}$ mutation, *ety13* animals underwent a second round of CRISPR injection, using the HPD crRNA TGTCGTCTGTATGCTTGTCG and the HPD > QPN repair template CAACTTTCAGGTACCGAAAATTAGCCTTAAAATGGCAACCGAACAAG CATACAGA CGACAAATCAAAAGAAGAAGCCGA. Underlined nucleotides are mutations introduced to change the HPD sequence (from **HPD to Q**PN), and are surrounded by two 35nt-long homology arms. Rol/dpy F1 progeny were screened by PCR to identify the *dnj-24(ety13ety15)* ΔH5+QPN (DNJ-24$^{Open-QPN}$) allele. Each animal was subjected to two diagnostic reactions: GCACCTAGAGAAGATTCTCCA-TACA + CGTCTGTATGCTTGTTCAGT, specific for the CRISPR *ety15* QPN allele, and GCACCTAGAGAAGATTCTCCATACA + CGTCTG TATGCTTGTCGGGG, specific for the WT allele. Animals positive for the QPN edit were homozygozed and backcrossed to dispose of the *dpy-10* allele.

## Protein expression and purification

DNAJB6$^{JD}$ and DNAJB6$^{JD-GF}$ constructs were expressed in BL21(DE3) cells (Novagen) fused to a sumo tag, and grown in LB or M9-minimal media supplemented with $^{15}$NH$_4$Cl and $^{13}$C-glucose at 37 °C until OD$_{600}$ of 0.8. Expression was induced by the addition of 1.0 mM IPTG and allowed to proceed overnight at 25 °C. Following harvesting, cells were lysed by French Press in 50 mM Tris, pH 8.0 with 300 mM KCl, 10 mM imidazole, and purified on a 5 mL HisTrap HP Ni-NTA column (GE Healthcare). The proteins were released from the column with 50 mM Tris, pH 8.0, 500 mM KCl, 250 mM imidazole elution buffer. The His$_6$-sumo tag was removed by an overnight cleavage with Ulp1 protease in 50 mM Hepes pH 7.0 and 300 mM KCl at 4 °C. The cleaved proteins were further separated from the proteases and the uncleaved protein fractions by reverse capture Ni-NTA and further purified on a HiLoad 16/600 Superdex 75 pg gel (GE Healthcare) filtration columns. All the purification steps were performed at 4 °C and the buffers were supplemented with a protease inhibitor cocktail (Sigma) and phenylmethylsulphonyl fluoride (Roche).

Full-length DNAJB6$^{WT}$ and variants fused to a sumo tag were grown in LB media at 37 °C until OD$_{600}$ of 0.8 and their expression was induced by the addition of 1.0 mM IPTG overnight at 25 °C. Following harvesting, cells were lysed by French Press in 50 mM Tris, pH 8.0 with 300 mM KCl, 10 mM imidazole, and purified on a 5 mL HisTrap HP Ni-NTA column (GE Healthcare). The proteins were released from the column with 50 mM Tris, pH 8.0, 500 mM KCl, 250 mM imidazole elution buffer. The His$_6$-sumo tag was removed by an overnight

cleavage with Ulp1 protease in 50 mM Hepes pH 8.0 and 300 mM KCl at 4 °C. The cleaved proteins were further separated from the proteases and the uncleaved protein fractions by reverse capture Ni-NTA. The proteins were further purified on a HiLoad 16/600 Superdex 200 pg gel (GE Healthcare) filtration columns.

HTT$_{Ex1}$-Q48 was expressed in BL21(DE3) cells (Novagen) and grown to OD 0.8 at 37 °C. Protein expression was induced by the addition of 1 mM IPTG and allowed to express for 4 h at 30 °C. After harvesting, the cells were resuspended in 50 mM NaH$_2$PO$_4$, 150 mM NaCl, 1 mM EDTA, pH 8.0 buffer supplemented with protease inhibitors cocktail (Roche) and lysed with French press. The lysate was loaded on a glutathione–agarose beads (Amersham Pharmacia) and the protein was eluted in 50 mM Tris-HCl pH 8.6, 150 mM NaCl, 1 mM EDTA and 20 mM reduced glutathione. The eluted protein was further purified HiLoad 16/600 Superdex 200 pg gel (GE Healthcare) filtration columns and stored in 50 mM Hepes pH 7.4, 150 mM KCl, and 1 mM DTT buffer.

His$_6$-MBP-tagged TDP-43 was expressed in BL21-CodonPlus cells (Novagen) and grown to OD 0.8 at 37 °C. Following lysis by French press the lysate in 50 mM Tris-HCl pH 8, 300 mM NaCl, 10 mM imidazole was loaded on a 5 mL HisTrap HP Ni-NTA column (GE Healthcare). The proteins were eluted from the column with 50 mM Tris, pH 8.0, 300 mM KCl, 250 mM imidazole buffer. The proteins were further purified on a HiLoad 16/600 Superdex 200 pg gel (GE Healthcare) filtration column and stored in 50 mM HEPES pH 7.4. 150 mM KCl buffer.

Hsp70 WT and variants were expressed and purified as described previously[50].

## NMR spectroscopy

All NMR experiments were carried out at 298 K on 14.1 T (600 MHz), 18.8 T (800 MHz), or 23.5 T (1000 MHz) Bruker spectrometers equipped with triple resonance single (z) or triple (x,y,z) gradient cryoprobes. The experiments were processed with Topspin 4.1 (Bruker) or NMRPipe[83] and analyzed with NMRFAM-SPARKY[84] and CcpNmrAnalysis[85].

Isotopically labeled proteins for NMR were grown in M9 H$_2$O or D$_2$O media supplemented with $^{15}$NH$_4$Cl (and $^{13}$C-glucose) as the sole nitrogen (and carbon) source.

DNAJB6 with selective $^{13}$CH$_3$-ILVM methyl labeling was grown in M9 D$_2$O media supplemented with $^{15}$NH$_4$C and [$^2$H,$^{12}$C]-glucose as the sole carbon source. Then, 60 mg/L of 4-$^{13}$C-α-keto-butyrate (Cambridge isotope laboratories−CDLM-7318), 80 mg/L of α-ketoisovaleric acid, sodium salt precursor (CDLM-7317-PK) and 100 mg/L of L-Methionine (CLM-206-PK) were added 1 h prior to protein induction to achieve selective $^{13}$CH$_3$-methyl labeling.

## Assignments of J-domain and DNAJB6$^{JD-GF}$ mutated constructs

Backbone $^1$H, $^{15}$N and $^{13}$C resonance assignments for DNAJB6$^{JD}$ and DNAJB6$^{JD-GF}$ F89I, F91I, A50V, P96R, and F100V mutants were carried out on 2–3 mM samples in 50 mM Hepes pH 7.4 buffer supplemented with 50 mM KCl, 0.03% NaN$_3$ and 10% D$_2$O buffer. Assignments were obtained by recording HNCACB, CBCA(CO)NH, and HN(CA)CO experiments[86] on a 600 MHz Bruker spectrometer. Unambiguous assignment of 98%, 95%, 95%, 95%, 95%, and 92% were achieved for DNAJB6$^{JD}$, DNAJB6$^{JD-GF}$ F89I, F91I, A50V, P96R, and F100V constructs, respectively. A 3D $^{15}$N-edited NOESY[87] was acquired on DNAJB6$^{JD-GF}$ and DNAJB6$^{JD-GF}$ F93L $^{15}$N-labeled samples to aid with transferring the assignments from the WT DNAJB6$^{JD-GF}$ construct (BMRB entry 30655). 3D $^{15}$N-edited NOESY (150 ms mixing time)[87], was also carried out on DNAJB6_F100V$^{JD-GF}$ in addition to the triple resonance experiments to validate the assignments.

## DNAJB6 JD CS-Rosetta structure calculations

Structural model of DNAJB6$^{JD}$ was generated using CS-Rosetta[88–90].

As a first step, the pick-fragments application from the CS-Rosetta toolbox 3.0 was used for fragment selection, based on the backbone chemical shift data[91]. Overall, 546 chemical shifts were used as input (71 $^{13}C_\alpha$ shifts, 68 $^{13}C_\beta$ shifts, 69 $^{13}C'$ shifts, 67 $^{15}N$ shifts, and 67 $^{1}H_N$ shifts). The fragment library containing 3- and 9-residue fragments was assembled by scoring both against a library of fragments with chemical shifts predicted from SPARTA+[92], and against chemical shifts predicted from secondary structure elements.

We then used Rosetta's Ab initio Relax protocol with the chemical shifts and 54 NOEs as inputs, to generate 250,000 starting models. The resulting lowest energy 500 models (by total score) were then subjected to local simultaneous refinement of backbone and side chain conformations (Rosetta-relax), incorporating 67 residual dipolar coupling measurements (RDC). Each such run generated 500 models, with the same constraints as the original ab initio modeling. The top 10 models of the resulting ~250,000 models (by total score) are shown in Fig. 2b of the main text. Models are available as Supplementary Data 1.

## NMR binding experiments

$^{1}H$-$^{15}N$ HSQC experiments were carried out on WT or mutant DNAJB6$^{JD-GF}$ constructs, in the absence and presence of $^{1}H$ Hsp70 to monitor binding. Hsp70 T204A variant, deficient in ATP hydrolysis was used in the reaction to maintain Hsp70 in the ATP-bound state. All experiments were acquired in the presence of an ATP regenerating system[55,56].

The concentration of the DNAJB6$^{JD-GF}$ constructs was kept at 200 μM, and the amount of the Hsp70 was changed according to its affinity for the binding partner; 400 μM (for DNAJB6$^{JD-GF}$ WT), 200 μM (for DNAJB6$^{JD-GF}$ F89I, and F91I mutants), 100 μM (for DNAJB6$^{JD-GF}$ F93L, E54A, and P96R mutants), 50 μM (for DNAJB6$^{JD-GF}$ A50V, F100V, and DNAJB6$^{JD}$).

All titration experiments were carried out in 50 mM Hepes pH 7.4 buffer supplemented with 50 mM KCl, 2 mM ATP, 0.03% NaN$_3$ and 10% D$_2$O buffer, supplemented with an ATP regenerating system.

$^{1}H$-$^{15}N$ HSQC-TROSY spectra were acquired at 298 K for each sample and intensity ratios (I/I$_0$) were calculated, where I and I$_0$ correspond to the peak intensity of the bound and free samples, respectively. I/I$_0$ that were smaller than one standard deviation from the mean were considered significant.

## NMR chemical shift perturbations

Chemical shift differences between DNAJB6$^{JD-GF}$ WT and seven LGMDD1 mutants were monitored by 2D $^{1}H$–$^{15}N$ HSQC with 200 μM of $^{1}H$, $^{15}N$-labeled samples for each variant in buffer containing 50 mM Hepes pH 7.0, 50 mM KCl, 0.03% NaN$_3$ and 10% D$_2$O. The CSPs were calculated from the relation $\Delta\delta = \sqrt{[(\Delta\delta_H)^{*2} + (\Delta\delta_N/5)^{*2}]}$, where $\Delta\delta_H$ and $\Delta\delta_N$ are $^{1}H$ proton and $^{15}N$ nitrogen shift changes between the chemical shifts of WT and mutant DNAJB6$^{JD-GF}$ residues.

Chemical shift differences between the full-length DNAJB6 WT and six LGMDD1 mutants were monitored by 2D $^{1}H$–$^{13}C$ HMQC methyl-TROSY experiments[93] measured on 200 μM of [$^{2}H$, $^{13}CH_3$]-selectively methyl labeled DNAJB6 samples in 50 mM Hepes pH 8.0, 50 mM KCl, 0.03% NaN$_3$ and 100% D$_2$O.

CSPs were calculated from the relation$-\Delta\delta = \sqrt{[(\Delta\delta_H/\alpha)^{*2} + (\Delta\delta_C/\beta)^{*2}]}$, where $\Delta\delta_{H(C)}$ is the shift change between methyl group $^{1}H$ ($^{13}C$) nuclei in apo and fully saturated forms of the protein, $\alpha$ ($\beta$) is one standard deviation of the methyl $^{1}H$ ($^{13}C$) chemical shifts (separate values of $\alpha$ ($\beta$) are used for different methyl groups), as tabulated in the Biological Magnetic Resonance Data Bank (www.bmrb.wisc.edu). CSPs greater than one standard deviation from the mean were considered significant.

The percentage of the free DNAJB6$^{JD-GF}$ population for each DNAJB6 variant was calculated from their linear correlations of $^{15}N$ and $^{1}H$ chemical shifts. Since plots of chemical shifts of the various DNAJB6 disease mutations are linear it is reasonable to assume a two-site

exchange mechanism between inhibited (docked) and free DNAJB6$^{JD-GF}$ states that is fast on the NMR chemical shift time scale. Thus, the observed linear correlations of $^{15}N$ and $^{1}H$ chemical shifts with different DNAJB6 LGMDD1 mutants can be explained by a rapid two-state interconversion between inhibited/free conformations of the DNAJB6$^{JD-GF}$. The fast equilibrium enables estimation of the relative populations of each state from the positions of all reporter cross-peaks relative to their inhibited (DNAJB6$^{JD-GF}$ WT) and free (DNAJB6$^{JD}$) reference points. In this limit, $p_{free}$ values can be calculated according to the relation[94]:

$$\delta_{mutant} = p_I\delta_I + p_F\delta_F \qquad (1)$$

where $\delta_{mutant}$ is the chemical shift of the amide probe in the mutant, $\delta_I$ and $\delta_F$ are the corresponding chemical shifts in the DNAJB6$^{JD-GF}$ WT (inhibited) and DNAJB6$^{1-96}$ (free) states and $p_I/p_F$ is the fractional population of the $I/F$ state ($p_F+p_I=1$).

Residues that are near the mutation itself were excluded for each variant.

## Hsp70 ATPase activity assay

Hsp70 phosphate-release rates after ATP hydrolysis were measured under steady-state conditions by monitoring the change in fluorescence of the phosphate-binding protein (PBP) A197C mutant, which was labeled at Cys197 using 7-diethylamino-3-[N-(4-maleimidoethyl) carbamoyl]coumarin (MDCC, CDX-D0198 from Adipogen). Fluorescence was measured in a Synergy H1 plate reader by exciting at 430 nm and measuring at 465 nm. All reactions contained PBP, 0.25 μM Hsp70 in 50 mM HEPES pH 8.0, 150 mM KCl, 10 mM MgCl$_2$ and 2 mM DTT, and 0.25 μM of DNAJB6 variants. After the plate was incubated at 37 °C for 10 min, the reactions were started by injection of ATP to a final concentration of 100 μM into each well. Wells were then read every 40 s for the first 20 min, and every 2 min for the next 40 min. For each plate, a series of five phosphate concentrations with PBP alone was measured to generate a calibration curve, which was used to correlate fluorescence to the concentration of the released phosphate. All ATPase assays were performed in triplicate.

## Fluorescence polarization assays

Steady-state equilibrium binding of DNAJB6 variants to Hsp70 chaperone was measured by fluorescence anisotropy using 100 nM of fluorescently tagged Hsp70 T204A,S494C-AF488 (C267A, C574A, C603A) protein and indicated concentration (0–1.0 mM) of full-length DNAJB6$^{WT}$ or mutants. The proteins were incubated for 20 min at 37 °C in 50 mM HEPES pH 8.0, 200 mM KCl, 10 mM MgCl, with 2 mM ATP. Data was acquired on a BioTek Synergy H1 plate reader in black, flat-bottomed 384-square-well plates. The excitation filter was centered on 485 nm with a bandwidth of 20 nm, and the emission filter was centered on 528 nm with a bandwidth of 25 nm. Data were fit to a one-site binding model using OriginPro.

## Aggregation prevention

**Huntington polyQ aggregation-prevention assay.** GST-tagged HTT$_{Ex1}$-Q48 (10 μM) was pre-incubated in the presence or absence of DNAJB6 (0.05, 0.25, 0.5, 0.8 and 1.6 μM) or DNAJB6 LGMDD1 mutants (0.25 and 0.8 μM) for 10 min at 37 °C. All proteins in the assay were buffer exchanged into the assay buffer (50 mM HEPES pH 8.0, 200 mM KCl, and 1 mM DTT). DNAJB6 P96R mutant was unstable at pH 8.0 and the aggregation-prevention assays for this variant were performed in 50 mM HEPES pH 7.0 and 200 mM KCl buffer. Thioflavin T (ThT; Sigma) at a final concentration of 10 μM was added and the aggregation was induced by the cleavage of the GST solubility tag with 0.1 μM 3 C protease. Aggregation reactions were run for 15 h at 37 °C with continuous shaking (500 rpm) and monitored by ThT fluorescence (excitation = 440 nm, emission = 485 nm), using an area scan mode

with a 3 × 3 matrix for each well. Black, flat-bottom, 96-well plates (Nunc) sealed with optical adhesive film (Applied Biosystems) were used. The experiments were conducted in triplicate and the mean ± standard deviation is reported.

**TDP-43 aggregation prevention.** MBP-tagged TDP-43 (10 μM) was incubated in the presence or absence of 5 μM DNAJB6 WT or LGMDD1 disease mutants for 10 min at 37 °C. The aggregation was induced by cleavage of the MBP solubility tag with 0.1 μM TEV protease and monitored at 37 °C by measuring light scattering at 360 nm as a function of time. All reactions were performed in 50 mM of HEPES pH 7.4 buffer supplemented with 100 mM of KCl and 1 mM DTT. Assays were acquired in an area scan mode with a 3 × 3 matrix for each well in clear, flat-bottom, 96-well plates (Nunc) sealed with optical adhesive film (Applied Biosystems). Assays were conducted in triplicate and the mean ± standard deviation is reported.

**Thermal melts.** Thermal melts were performed with a nanoDSF instrument (NanoTemper), which was used to monitor the intrinsic fluorescence of full-length DNAJB6 variants as a function of temperature. Capillaries contained ~10 μL of each protein in a 50 mM Hepes pH 7.0 buffer supplemented with 150 mM NaCl and 2.5 mM DTT. The initial temperature was 20 °C and was set to increase by 1 °C per minute. Fluorescence readings were recorded at 330 and 350 nm, and the melting temperature ($T_m$) was extrapolated from the inflection point.

**SEC-MALS.** Solutions of DNAJB6 WT, LGMDD1 disease mutants, or DNAJB6 ΔST (100 μM) were separated by size exclusion chromatography on a Superose 6 Increase 10/300 GL column. Molecular weights were determined by multi-angle laser light scattering using an in-line DAWN HELEOS detector and an Optilab T-rEX differential refractive index detector (Wyatt Technology Corporation). Calculation of molecular weights was performed using the ASTRA software package (Wyatt Technology Corporation).

**Overexpression of *hsp-1* in *C. elegans* muscles**
*hsp-1* cDNA was amplified from a *C. elegans* cDNA library using primers cataacatagaacattttcagGAGGAATGAGTAAGCATAACGCTGTTGG and CGACCGGCGCTCAGTTGGAATTCTAGATTAGTCGACCTCCTCGATCG, and cloned (NEBuilder kit) into a plasmid backbone that carries the *myo-3p* muscle-specific promoter to generate a *myo-3p::hsp-1* transgene that was injected together with a *myo-2p::mCherry* co-marker (labels the pharynx) into WT or *dnj-24(ety13)* animals.

**Western blots**
Worms were lysed on ice using glass beads in lysis buffer (50 mM HEPES, pH 8.0, 500 mM KCl, 2 mM MgCl₂, 0.1 mM ETDA, 0.5 mM EGTA-KOH, 15% glycerol, 0.1% NP-40) supplemented with protease inhibitor cocktail (Roche). Protein concentrations were determined using a BCA Protein Assay Kit (Thermo Fisher Scientific). Aliquots of lysates were solubilized in Laemmli sample buffer, and equal amounts of proteins were separated on 4–20% gradient SDS-PAGE gels (GenScript). Proteins were transferred onto nitrocellulose membranes and then blocked with 5% nonfat dry milk in TBS buffer for 1 h. The membranes were then incubated with primary antibodies, specific to the protein of interest, in TBS overnight at 4 °C. After incubation with the appropriate secondary antibody conjugated with HRP, ECL (Clarity, Bio-Rad) was used for protein detection. Immunoblots were obtained using the ChemiDoc MP Imaging System (Bio-Rad). Densitometry was measured with ImageJ software (NIH).

The following antibodies were used: Hsp70 (3A3, ab5439, Abcam, 1:5000), DNAJB6 (B-5, sc-365574, Santa Cruz, 1:1000), GAPDH (G8795, Sigma-Aldrich, 1:1000), goat anti-mouse IgG-HRP (JIR 115-035-003), and goat anti-rabbit IgG-HRP (JIR 111-035-003).

**Phalloidin staining of actin filaments for visualization of muscle fibers**
Phalloidin staining was done as previously described[95]. Briefly, worms were washed off agar plates, snap-frozen in liquid nitrogen and permeabilized using ice-cold acetone. Worms were stained with Phalloidin-FITC (2.5 U per replica, Sigma) and mounted on an agarose pad for imaging. A Zeiss LSM 880 confocal microscope was used with 63× or 40× magnification, Zen software was used for imaging (version 2.3). All the statistical analyses were computed using GraphPad version 9.

**Thrashing assay**
Twenty-four one-day adult worms of each genotype were singly transferred to 24 wells of a 96-well plate filled with 200 μL M9 buffer and after 10 min of habituation, thrashing was recorded for 30 s using a Kastl automated observation box (LoopBio, Vienna). Recordings were converted to MP4 files using the Motif software and then analyzed visually for the number of thrashes using VLC software run at 12x slower speed.

**Reporting summary**
Further information on research design is available in the Nature Portfolio Reporting Summary linked to this article.

## Data availability
NMR chemical shifts have been deposited in the Biological Magnetic Resonance Data Bank (BMRB) under the following accession codes: 51996 for DnaJB6$^{JD}$, 52098 for DNAJB6$^{JD-GF}$ F100V, 51997 for DNAJB6$^{JD-GF}$ A50V, 51998 for DNAJB6$^{JD-GF}$ P96R, 52091 for DNAJB6$^{JD-GF}$ F91I, and 52065 for DNAJB6$^{JD-GF}$ F89I. The protein structure of DNAJB6 used for analysis in this study is available in the Protein Data Bank under accession code 6U3R. Data supporting the findings of this study are available in the Supplementary Information file. CS-Rosetta models of the DNAJB6$^{JD}$ generated as part of this study are provided in Supplementary Data 1. Source data are provided with this paper.

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

## Acknowledgements
The authors would like to thank T. Scherf for NMR support and the Clore Institute for High-Field Magnetic Resonance Imaging and Spectroscopy. We also thank D. Fass, M.P. Latham, and O. Faust for helpful discussions and advice. R.R. is supported by the European Research Council (ERC-2018-STG 802001), the Israel Science Foundation grant 1093/22, the Minerva Foundation, the Abisch-Frenkel Foundation for the Promotion of Life Sciences, the Helen and Martin Kimmel Institute for Magnetic Resonance Research, and the Blythe Brenden-Mann New Scientist Fund. M.O.-S. acknowledges financial support from the European Research Council (ERC-2019-STG 850784), Israel Science Foundation grant 961/21, Dr. Barry Sherman Institute for Medicinal Chemistry, Sagol Weizmann-MIT Bridge Program, and the Azrieli Foundation.

## Author contributions
M.A.-A., Y.S., D.G., M.O.-S., and R.R. conceived and designed the experiments and analyzed data. M.A.-A., D.G., and R.R. recorded and analyzed the NMR data. M.A.-A. and R.R. performed the biochemical and functional assays. Y.S. and M.O.-S. performed the *C. elegans* studies. M.A.-A. and R.R. wrote the initial draft and all authors have reviewed and edited the manuscript.

## Competing interests
The authors declare no competing interests.
