## [Peer Review File · Nature Communications]

REVIEWER COMMENTS

Reviewer #1 (Remarks to the Author):

The manuscript by Avraham et al., is interesting, generally well written and the experiments are designed properly. The generation of an in vivo model from structural data is perhaps one of the strongest aspects of this work. Although I generally avoid making comments about novelty, I note that the results presented in the manuscript from X et al., are to a large degree expected from data already published in the literature (refs 6, 14, 15, 19, 35) and also Bhadra Nat Comms 2020. These studies showed that the mutations alter the interaction of Hsp70 with Hsp40 and also revealed the presence of a regulatory helix in DNAJB6. Based on the location of the mutations in relation to the regulatory helix it is straightforward to deduct that the regulation of the HSP70 cycle is altered in the mutants (for instance see figure 4 in ref 6). The results in this manuscript essentially confirm this hypothesis and therefore paragraph 3 in the introduction is not strictly true. That does not mean that the data presented in the manuscript are not worthy of publication, especially since the manuscript unifies a lot of the proposed mechanisms. Nevertheless, the authors could do a much better job in placing their findings in the context of already existing data/hypothesis in the introduction and throughout the paper. Aside from this general comment I have some more specific questions below:

1. C.elegans model:

a) This to me is the most interesting part of the paper. However, I was confused as to why the authors did not test the LGMDD mutants in their worm model and instead chose to use an artificial DNAJB6-open mutation. If the A50V and F100V mutations cause a complete release of autoinhibition, should they not behave like the open mutant? These experiments will provide much stronger evidence for the effect of the mutations in vivo.

b) A better justification of why dnj-24 and not any of the other class-B DNAJs was chosen needs to be provided, maybe a sequence alignment of the full-length proteins?

2. Anti-aggregation function:

The data presented in Figure 1 are very clear-cut and conclusively show that the mutations do not affect anti-aggregation in vitro. However, as the authors mention, LGMDD mutations are associated with a decreased anti-aggregation activity in vivo (refs 13, 16). In fact, the Hsp70-independent anti-aggregation function of DNAJB6 has come under scrutiny, see for instance ref 32. This suggests that in vivo, anti-aggregation might be taking place through a different mechanism than what happens in vitro. The authors provide a reference of (Adupa et al. 2023) to say that this is not true, but I am not sure what this refers to. I do agree that the C.elegans data provide strong evidence for a gain of function mechanism, but can we completely exclude that the mutations cause a loss of DNAJB6 function? i.e that the direct

DNAJB6-Hsp70 cooperation is affected by the LGMDD mutations in vivo and that leads to increased aggregation?

3. NMR studies

a) It is not clear to me how the authors arrive to the populations shown in Figure 3. Data from different residues point to different populations of the open state. As the authors mention, 21 residues show a linear behaviour in their peak positions but only 4 are shown in Figure 3. Do the rest behave the same? Data from all residues should be shown as supplementary information. In general, the statement about the mutations progressively releasing autoinhibition seems too strong in the absence of crystal structures (which I understand would not be possible in such a dynamic system). I would advise to tone down the language to something like 'mutations shift DNAJB6 to an Hsp70 binding-competent conformation'.

b) Am I right to think that the helix 5 closed – open transition happens fast ($\sim 2500 \text{ s}^{-1}$) leading to the CSPs shown in Figure 3 but Hsp70 binding must be a lot slower to produce the changes in intensity shown in Figure 4? If that is the case the how can two events taking place in such different timescales affect each other? Are there any CSPs upon addition of Hsp70 or only intensity changes? In figure 4 there is little correlation between the population of the open state and the ability to bind Hsp70, but the ATPase data nicely reflect the increased availability of the J domain in the mutants.

c) The section about 'full-length' DNAJB6 in page 11 is partially misleading. The NMR data shown on DNAJB6-mono are collected on a construct that lacks the ST domain (~ 50 residues, that were implicated in substrate binding) so that is definitely not full-length. This has to be corrected in the text, supplementary and figures to make it clear. This construct must be $\sim 25 \text{ kDa}$ and therefore it should be amenable to traditional ^1H ^{15}N HSQC spectra. Do these data agree with the methyl NMR data shown in figure 5 and S7? Does the methyl and ^{15}N spectrum of DNAJB6-mono overlay with the spectrum of DNAJB6-JD-GF?

d) Previous work from the Rosenzweig lab on DNAJB1 suggested that transient interactions between JD and CTD play a role in autoinhibition release. Is that the case for DNAJB6? This needs to be addressed in the discussion.

4. All assigned chemical shifts for all mutants should be deposited in the appropriate database.

5. Figure 6 legend, parts a and b seem mixed up. Error bars in panels e and g?

6. Some referencing issues throughout.

Reviewer #2 (Remarks to the Author):

The manuscript describes a set of studies that significantly advance our understanding of the mechanisms underlying mutations in DNAJB6 that are associated with a LGMD. The data is of high quality and the conclusions drawn from the data are largely convincing (but see points below).

The main novelties of the work are the findings that mutations in DNAJB6 JD and G/F region linked to LGMD1 likely dysregulate binding of the JD to Hsp70 (mutations release resting state inhibition of the binding interaction), and that the dysregulation leads to sequestering of Hsp70s away from other roles in the cell. The G/F helix docked to the J-domain in DNAJB6 was shown previously in the structure determined by Karamanos et al., and appears to be operate analogously to what was seen in DNAJB1. The results suggest a simple correlation between helix V unbinding to the JD and binding of Hsp70 to the JD, and this correlation extends also to ATPase stimulation. It is not obvious however why some of these mutations provide more or less unbinding between helix V and the JD in the first place. In addition, a potential novelty with regard to mechanism could have been in how client proteins induce release of the J-domain inhibition (which the authors point out must be different to that of DNAJB1) but that is not addressed here.

Questions:

The mechanism by which some of the mutations affect helix V dissociation from the JD is not obvious from the residue positions and the structures shown. Some discussion of this and perhaps more analysis of the mutation sites seems appropriate. What mutation sites make direct contact with the JD, and is there any evidence for conformational flexibility not reflected in the static structures that might allow for close contacts that otherwise appear long-range in the models? Some of these substitutions are in what could be flexible loops, but have dynamics been looked at? The chemical shift perturbation data of Supp Fig 2 seems like it could be a rich source of information for understanding long range effects from the mutations.

The proposal is for a simple link between helix V undocking and Hsp70 binding. Flipping back and forth between the helix V docking percentages shown in Fig. 3(c) and the ATPase activities shown in Fig. 4 suggests that the correlation will be good but I suggest that a correlation plot be presented. A correlation plot provides a quantitative measure, and also reveals any nonlinearity or outliers, which might help formulate additional mechanistic hypotheses (for example, perhaps pointing to mechanisms

related to the variable changes in helix 5 intensity loss noted below). The correlation with ATPase activities for full-length DNAJB6 from Fig. 6(a) should be assessed similarly.

The helix V intensity losses for the various mutants on binding Hsp70 are surprising and unexpected for a simple model in which helix V undocks from the JD and provides access for Hsp70 to bind the JD. The unexpected intensity losses are noted by the authors but there aren't any real insights as to what is happening here. There is not an obvious trend across the mutants that correlates with the severity of the undocking of helix V, and a more or less rigid link between helix V and the JD seems unlikely. There could be a combination of factors, including how the mutation affects the ability of the helix V to get out of the way of Hsp70, subtle changes in pose, whether there are nonspecific contacts with the Hsp70 and perhaps chemical exchange processes, etc. If the correlation between (un)docking percentage and Hsp70 binding/ATPase activity is very high, then an in depth analysis of this is probably not very interesting, but it nonetheless begs questions about what is actually known about the Hsp70/JD complex interface and what restrictions Hsp70 binding imposes on where helix V can be positioned. If there are reliable models for any Hsp70/JD complex then perhaps this should be provided and discussed. It may be worth noting that the AlphaFold2-multimer, which produces a good quality pose with low PAE for the JD (full-length Hsp70 required), places helix V in the SBD binding site. Of course, the relevance of this is unknown but could be tested with SBD mutants.

There are some comments/concerns about the assessment of DNAJB6 anti-aggregation activity with and without Hsp70 (end of 2nd paragraph in Results, and Supp Fig. 1(b)), although it's not clear that the data here is actually necessary since the conclusion from the experiments is that "DNAJB6 chaperones can operate in an Hsp70-independent manner", which can be determined from the assays in the absence of Hsp70. In any case: Hageman & Kampinga et al. 2010 (ref 31) observed a similar result in cells (which should be cited and referred to here directly), and went further by showing that the J-domain (and so the Hsp70 interactions) of DNAJB6 (or DNAJB8) did not significantly aid in polyQ clearance for polyQ74, but did help against polyQ119. This suggests that it depends on the polyQ length. The manuscript results are for a polyQ48, which means this is not expected to be a stringent test of whether Hsp70 helps or not. Presumably any boost provided by Hsp70 would also depend on the relative concentrations, but I can't see that it is indicated what were the concentrations of either Hsp70 or DNAJB6 in the fibrilization assays of Supp. Fig 1(b). It should be noted however that the Hageman et al. results suggest that the additional activity provided by Hsp70s was due to the ability of Hsp70s to pass the polyQ to the cellular degradation machinery, which of course is not possible in this in vitro set-up. Finally, to observe a holdase activity of Hsp70, would it be necessary to include a NEF? In the canonical Hsp70 cycle, ATP hydrolysis stimulated by a J-domain induces substrate binding to the Hsp70, but release of substrate requires nucleotide exchange, which can be slow. Thus there are two possibilities that are difficult to assess here: Is it possible that Hsp70 is not turning over and becomes stalled in the absence of a NEF? Or is polyQ48 below the length and/or at a concentration below the capacity of DNAJB6? Any of these results seem possible here, and I can't see that the conclusion that Hsp70 provides no additional anti-aggregation activity is either accurate or particularly useful (or necessary for the conclusions?).

The authors state with regard to the data in Supp. Fig 5: "No changes were detected in other regions of the protein, strengthening our observation that client binding of DNAJB6 is not affected by the pathogenic mutations". I suggest this statement be removed since the data cited here is on a construct that is missing the ST region, which is known to be involved either directly or indirectly in substrate binding, as shown in Fig. 1(d) and 1(e). The data in Fig. 1(d) and 1(e) is a better test of whether the mutations affect substrate binding.

The work in *C. elegans* is very nice and a strong component of the manuscript. The high expression level of DNAJB6 in worm muscle cells is shown and also Meissner et al. 2011 is cited. The high DNAJB6 expression level is important for the hypothesis of Hsp70 sequestering. Is it known whether DNAJB6 is also highly expressed in human muscle cells? If it is known to be the case, then it should be cited, and if not or it is not known, then the discussion should be modified to reflect that.

More minor comments:

Nomenclature of constructs is often confusing:

- It should be stated clearly in Materials and Methods that constructs are based on human isoform b of DNAJB6.
- The use of the label "WT" is used in figures whether it is full-length or the JD-GF construct. It should be made clear in all figures and legends which construct (full-length or truncated) is being used.
- It seems the JD+GF construct is referred to as either DNAJB6(GF) or DNAJB6(JD-GF). There are many occurrences of both – I've assumed there is no data presented for a GF-only construct.
- Similarly, the name given to the "monomeric WT DNAJB6" should more clearly indicate that there is an internal deletion and so not wildtype throughout (e.g., DNAJB6(mon)). This same construct seems to be sometimes referred to as DNAJB6 deltaST as well, but in some cases it's not really possible to be sure of this from the way they are presented and discussed. It's appreciated that the WT is in reference to the sequence in the JD/GF region but it is confusing without having it identified as full-length vs. truncation.

There are quite a few errors/omissions/typos:

- Fig 1(d) and 1(e), it should be indicated what are the DNAJB6 and Q48/TDP-43 concentrations used in these t1/2 measurements.
- The coordinates for the structures shown in Fig 1(b) and Fig 3(a) appears to come from 6U3R (or 6U3S), rather than the 7JSQ indicated in the legends.
- Fig 3(a), would be good to have helices numbered to help with following the text. In addition, the location of the HPD motif that is primarily responsible for Hsp70 interactions should be shown.

- In Assignments section of Materials & Methods: what sites are being referred to when indicating the "unambiguous assignment" of the constructs? Is this backbone amide or does this include backbone heavy atom or other protons, etc.?

- Fig. 5(a) & 5(b), and Supp Fig. 5(a): Is the text embedded in the spectra indicating the ¹³C methyl labelling correct? This seems to be the only mention of an "LV" labelled sample, whereas all the other methyl-labelled samples are "ILVM".

- Fig. 1(a), the DNAJB6 protein should be 241 amino acids long, not 240?

- Fig. 5 legend 2nd to last sentence: "while the A50V mutant enhances activity 6.5" presumably should end with "...6.7-fold"?

- Fig. 6 legend, (a) and (b) descriptions appear to be swapped.

- Fig. 6(e) and 6(g), should have error bars.

- Residue/methyl numbering should be indicated in Figs. 5(c) & 5(d).

- Residue/methyl numbering should be indicated in Supp. Figs. 4(f) & 4(g).

- Discussion, 2nd to last paragraph, the call (DNAJB6 expression levels) should be to Supp Fig 6e.

Reviewer #3 (Remarks to the Author):

The authors nicely characterize the effects of common mutations causing LGMD1 on the structure and two molecular functions of the mutated protein DNAB6 in vitro. In doing so they provide a unifying theory of the commonality of the disease causing mutations, specifically that they impact HSP70 function in a dominant interfering fashion. The dominant fashion is particularly important because this is an autosomal dominant disease and therefore loss of DNAB6 function would not be expected to produce a dominant disease (indeed knockdown the *C. elegans* orthologue appears to produce a sterile or lethal phenotype and the mouse knockout is also embryonic lethal). Having nicely demonstrated a unifying theory for how the mutations might act at the molecular level in vitro, the authors introduce one of these mutations into *C. elegans* and find that it does indeed produce dystrophic muscles (note they did not demonstrate that it does so in a dominant fashion). The authors are successful at partially rescuing the dystrophic phenotype by introducing a second site mutation in DNAB6/DNJ-24, suggesting that at least part of the mechanism underlying the dystrophic phenotype is due to the interaction with HSP70. The authors further provide evidence for the possible HSP70 sequestration mechanism by overexpressing HSP70/HSP-1 in the presence of the mutant form of DNAB6/DNJ-24.

- What are the noteworthy results?

The most noteworthy results are: a) demonstration of a dominant mechanism of action underlying a dominant disease (this is important as much of the field has been focused on loss of function mechanisms); b) demonstration of the feasibility to develop a rapid model (*C. elegans*) for increasing mechanistic understanding and treatment of LGMDD1.

- Will the work be of significance to the field and related fields? How does it compare to the established literature? If the work is not original, please provide relevant references.

The work is highly significant to the field as it is paradigm shifting. The work will also be of interest to those studying proteostasis (including diseases), muscle physiology, and loss of muscle homeostasis with age. Compared to past literature this work is focused on incorporating personalized patient data into mechanistic understanding. Such work has been useful particularly for CFTR and is increasingly being looked at in rare diseases such as LGMDD1.

- Does the work support the conclusions and claims, or is additional evidence needed?

The work broadly supports the conclusions and claims. Unsurprisingly, the main areas for potential improvement relate to the *in vivo* experiments which are “proof of principle” at the end of the manuscript. For example, only phalloidin staining is used to look at actin structure, otherwise the muscle histology/physiology is not examined. To be fair, this would be a paper in and of itself (movement (and effect of interventions), other structure w/wo interventions (myosin, Z lines, M lines, mitochondria), Ca⁺⁺ levels w/wo interventions (see <https://pubmed.ncbi.nlm.nih.gov/36935171/>), PolyQ/other aggregation w/wo interventions, and possibly EM). Thus, I would encourage the authors to consider these experiments and to publish a follow on paper or papers having apparently established a novel model for LGMDD1 that could then be exploited for therapeutic discovery (and more mechanistic studies). If I were to ask for additional experiments, it would probably just be movement data to go with Figure 6d and f (e.g. thrash assays in these 4 conditions) assuming these worms are currently in the lab this would be 1-2 weeks of work.

Other thoughts: 1) It might be reasonable to suggest that other proteins could also be binding to DNAJB6 and therefore not everything has to be dependent upon HSP70 (even if it is likely the predominant mechanism). 2) It might be reasonable to consider clinical data (for example <https://pubmed.ncbi.nlm.nih.gov/36427278/>) vs. your unifying model in the discussion section (e.g. the model doesn't immediately fit the clinical data, of course this could be due to individual genomic differences as well)

- Are there any flaws in the data analysis, interpretation and conclusions? - Do these prohibit publication or require revision?

The main reservations I have are with potentially overstating things. For example “unregulated” vs. “dysregulated” and “Excitingly” and “conclusively” and “disease phenotype” vs. potentially multiple phenotypes with potentially multiple mechanisms.

The second reservation I have is with suggesting the protein aggregation is not altered in vivo (I think this is a wording issue confusing DNAJB6 direct chaperone activity vs. direct HSP70 chaperone activity), esp. as it is part of the clinical phenotype (and probably is occurring based upon hsp-1 knockdown results in wormbase <https://pubmed.ncbi.nlm.nih.gov/15084750/>). I raise this as the phenotype of hsp-1 includes increased protein aggregation (and a movement defect and altered mitochondrial structure) amongst other things (for example probably altered stress responsiveness based upon impacts on daf-16 localization). Additionally, you’ve not assessed protein aggregation in vivo thus it might be better to tone down the language around this a bit.

Lastly, the images of Open in 6 d vs f display quite distinct pathologies. D is reminiscent of DMD (<https://pubmed.ncbi.nlm.nih.gov/11696327/>) whereas F is reminiscent of tropomodulin mutants/nemaline pathologies (<https://pubmed.ncbi.nlm.nih.gov/17976644/>) which present more severely and are associated with protein aggregation and defects more at the myotendinous junction than the Z/M line. It might be worth either subclassifying minor vs. severe (in e and g graphs) or displaying consistent minor images in d and f.

- Is the methodology sound? Does the work meet the expected standards in your field?

Yes and yes.

- Is there enough detail provided in the methods for the work to be reproduced?

Yes.

Reviewer #1 (Remarks to the Author):

The manuscript by Avraham et al., is interesting, generally well written and the experiments are designed properly. The generation of an in vivo model from structural data is perhaps one of the strongest aspects of this work. Although I generally avoid making comments about novelty, I note that the results presented in the manuscript from X et al., are to a large degree expected from data already published in the literature (refs 6, 14, 15, 19, 35) and also Bhadra Nat Comms 2020. These studies showed that the mutations alter the interaction of Hsp70 with Hsp40 and also revealed the presence of a regulatory helix in DNAJB6. Based on the location of the mutations in relation to the regulatory helix it is straightforward to deduct that the regulation of the HSP70 cycle is altered in the mutants (for instance see figure 4 in ref 6). The results in this manuscript essentially confirm this hypothesis and therefore paragraph 3 in the introduction is not strictly true. That does not mean that the data presented in the manuscript are not worthy of publication, especially since the manuscript unifies a lot of the proposed mechanisms. Nevertheless, the authors could do a much better job in placing their findings in the context of already existing data/hypothesis in the introduction and throughout the paper.

We have modified the introduction section to address the points made by the reviewer.

“To date, 16 pathogenic mutations have been identified in DNAJB6^{13,15,16}, however the mechanism by which these mutations affect the structure and function of the chaperone, leading to LGMDD1 disease, is not fully understood.”

It is important however to note that while the results may be “to a large degree expected”, there is currently no study characterizing the structure of LGMDD1 mutants or their mechanism of function. Furthermore, while the seminal study by Karamanos et al (ref 19) discovered the presence of the inhibitory helix in DNAJB6, there has been no experimental data showing that this helix indeed blocks or regulates Hsp70 binding.

Generally, I find it is important to insist on also exploring things experimentally, as many times the results surprise us, with the actual functional mechanisms turning out to be very different from the “expected” ones.

Aside from this general comment I have some more specific questions below:

1. C.elegans model:

a) This to me is the most interesting part of the paper. However, I was confused as to why the authors did not test the LGMDD mutants in their worm model and instead chose to use an artificial DNAJB6-open mutation. If the A50V and F100V mutations cause a complete release of autoinhibition, should they not behave like the open mutant? These experiments will provide much stronger evidence for the effect of the mutations in vivo.

We thank the reviewer for this positive comment. We specifically decided to use an artificially open mutation and not the disease mutant to show that it is the loss of inhibition itself that causes the disease phenotype. The introduction of LGMDD1 mutations into mice and zebrafish has been previously shown to cause defects in muscle morphology (Bengoechea et al (ref 64) and Nam et

al (red 65)). We, however, felt that showing that the artificial release of the inhibition (without affecting any of the specific residues associated with the disease) would be much stronger proof for our proposed mechanism - where it is the loss of inhibition and the uncontrolled recruitment of Hsp70 chaperones that is the cause for the disease.

We have changed the relevant text in the revised manuscript to make this point clearer.

“An artificially open DNJ-24 mutant (DNAJB6^{Open}), lacking JD-GF inhibition, was designed by substituting four conserved residues in GF helix V with glycines or serines (Fig. 6c and Supplementary Fig. 9d) in dnj-24 using CRISPR-Cas9. These mutations were specifically designed so they did not affect the residues found to be mutated in LGMDD1 patients, and thus any effects could be attributed solely to the disruption of the interaction between GF helix V and the J-domain.”

b) A better justification of why dnj-24 and not any of the other class-B DNAJs was chosen needs to be provided, maybe a sequence alignment of the full-length proteins?

The reason for choosing *dnj-24* is that it is the only DNAJB6 homolog present in *C. elegans*. DNAJB6 belongs to the non-canonical class B JDPs. There are 4 in humans and only one in *C. elegans*.

This is further explained in the text of the revised manuscript: *“ Furthermore, while there are four non-canonical human class B JDPs, DNJ-24 represents the only member of this family found in nematodes.”*

2. Anti-aggregation function:

The data presented in Figure 1 are very clear-cut and conclusively show that the mutations do not affect anti-aggregation in vitro. However, as the authors mention, LGMDD mutations are associated with a decreased anti-aggregation activity in vivo (refs 13, 16). In fact, the Hsp70-independent anti-aggregation function of DNAJB6 has come under scrutiny, see for instance ref 32. This suggests that in vivo, anti-aggregation might be taking place through a different mechanism than what happens in vitro. The authors provide a reference of (Adupa et al. 2023) to say that this is not true, but I am not sure what this refers to. I do agree that the *C. elegans* data provide strong evidence for a gain of function mechanism, but can we completely exclude that the mutations cause a loss of DNAJB6 function? i.e. that the direct DNAJB6-Hsp70 cooperation is affected by the LGMDD mutations in vivo and that leads to increased aggregation?

We thank the reviewer for raising this important point. Based on our in vivo data, we indeed proposed that the reduced anti-aggregation effect seen with the LGMDD1 DNAJB6 mutants is due to the unregulated interaction with Hsp70 and not due to the loss of DNAJB6 chaperoning activity.

If the chaperoning activity of the DNAJB6 mutants was affected by the mutations, overexpression of Hsp70 and /or eliminating the Hsp70 interaction via the HPD mutation would not be able to suppress the phenotype.

It is important to note that we do not claim that the aggregation-prevention activity of DNAJB6 is entirely Hsp70-independent in the cell, but rather that the mutations do not affect the chaperoning activity of DNAJB6 itself in aggregation prevention. Thus the decreased anti-aggregation activity of DNAJB6 mutants observed in cells is due to the unregulated interaction of these mutants with Hsp70, causing the toxic gain of function through Hsp70 depletion.

We have revised the relevant discussion section to make this point clearer

Our findings indicate that this high-affinity, non-productive interaction with mutant DNAJB6 depletes cellular levels of Hsp70 chaperone, disrupting protein homeostasis and leading to aggregation. Therefore, it is not, as previously suggested, the loss-of-function of DNAJB6 that underlies the disease, but rather the unregulated binding of DNAJB6 LGMDD1 mutants to Hsp70. This hypothesis is further supported by recent findings demonstrating that inhibitors of the JD-Hsp70 interaction can reduce the severity of LGMDD1 disease³⁵, and explains previous observations that overexpression of WT DNAJB6 has no corrective effect on muscle morphology^{13,35}.

3.NMR studies

a) It is not clear to me how the authors arrive to the populations shown in Figure 3. Data from different residues point to different populations of the open state. As the authors mention, 21 residues show a linear behaviour in their peak positions but only 4 are shown in Figure 3. Do the rest behave the same? Data from all residues should be shown as supplementary information. In general, the statement about the mutations progressively releasing autoinhibition seems too strong in the absence of crystal structures (which I understand would not be possible in such a dynamic system). I would advise to tone down the language to something like 'mutations shift DNAJB6 to an Hsp70 binding-competent conformation'.

We thank the reviewer for bringing this up, and realize that the description of this calculation was mistakenly omitted from the material and methods section.

We find that all the residues in the protein that show different chemical shifts between the docked (WT) and the undocked (DNAJB6-JD¹⁻⁹⁶, lacking helix5) conformations display this linear behavior. Thus, the observed linear correlations of ¹⁵N and ¹H chemical shifts with different DNAJB6 LGMDD1 mutants can be explained by a rapid two-state interconversion between inhibited/free conformations of the DNAJB6^{JD-GF}. The fast equilibrium enables estimation of the relative populations of each state from the positions of all reporter cross-peaks relative to their inhibited (DNAJB6^{JD-GF} WT) and free (DNAJB6^{JD}) reference points. In this limit, p_{free} values can be calculated according to the relation (Palmer 2001, reference 91) :

$$(1) \delta_{mutant} = p_I * \delta_I + p_F * \delta_F$$

where δ_{mutant} is the chemical shift of the amide probe in the mutant, δ_I and δ_F are the corresponding chemical shifts in the DNAJB6^{JD-GF} WT (inhibited) and DNAJB6^{JD} (free) states, and p_I/p_F is the fractional population of the *I/F* state ($p_I+p_F=1$).

Residues that are near the mutation itself were excluded for each variant.

A detailed explanation for this calculation was added to the materials and methods of the revised manuscript and, as requested by the reviewer, we have generated a new supplementary figure 3a showing the linear chemical shift behavior for all 21 residues.

b) Am I right to think that the helix 5 closed – open transition happens fast ($\sim 2500 \text{ s}^{-1}$) leading to the CSPs shown in Figure 3 but Hsp70 binding must be a lot slower to produce the changes in intensity shown in Figure 4? If that is the case the how can two events taking place in such different timescales affect each other? Are there any CSPs upon addition of Hsp70 or only intensity changes? In figure 4 there is little correlation between the population of the open state and the ability to bind Hsp70, but the ATPase data nicely reflect the increased availability of the J domain in the mutants.

The docking/undocking of the inhibitory helix indeed occurs through a rapid two-state interconversion [$>2500 \text{ s}^{-1}$] between inhibited and free conformations of the DNAJB6^{JD-GF}, while the binding of Hsp70 is a slower process occurring on a ms time scale as evident by the peak broadening in our spectrum. Due to the very large size of Hsp70 (70 kDa) and the on/off rates of the interaction, the binding of Hsp70 only caused changes in intensity, and we could not detect CSPs.

Since the inhibited/free transition is much faster than binding, it reaches a pre-equilibrium on the timescale of binding. Which means that the thermodynamics of the docking/undocking dominates the reaction and the flux is only governed by the relative fraction of the free DNAJB6^{JD-GF} (p_{free}).

The reviewer rightfully points out that this should result in a linear correlation between the population of the free JD-GF in each variant and its binding and cavitation of Hsp70. We indeed observe very good correlation for these two activities and have added a correlation plot for Hsp70 activation by DNAJB6 JD-GF variants to Supplementary Fig. 3b of the revised manuscript.

In the case of figure 4, the binding experiments between DNAJB6^{JD-GF} variants and Hsp70 were performed with different Hsp70 concentrations for each mutant to allow us to map the sites of Hsp70 binding even for the mutants with low population of the free state and thus lower affinity for Hsp70. This is the reason that a direct correlation can not be seen there. There is, however, a clear inverse correlation between the amount of Hsp70 that was required to detect significant peak broadening in our NMR spectrum and the fraction of the free JD-GF conformation for each DNAJB6^{JD-GF} variant. Higher concentrations of Hsp70 were added to the mutants with low populations of the free JD-GF (2-fold excess of F89I). While much lower Hsp70 concentrations were used for the more open mutants (0.25-fold excess for A50V and F100V mutants).

We have modified Figure 4 and the legend to clearly indicate that different concentrations of Hsp70 were used to generate the plots.

c) The section about 'full-length' DNAJB6 in page 11 is partially misleading. The NMR data shown on DNAJB6-mono are collected on a construct that lacks the ST domain (~50 residues, that were implicated in substrate binding) so that is definitely not full-length. This has to be corrected in the text, supplementary and figures to make it clear. This construct must be ~25 kDa and therefore it should be amenable to traditional ¹H ¹⁵N HSQC spectra. Do these data agree with the methyl NMR data shown in figure 5 and S7? Does the methyl and ¹⁵N spectrum of DNAJB6-mono overlay with the spectrum of DNAJB6-JD-GF?

We thank the reviewer for pointing this out and now provide a clear indication in the text and materials and methods that the monomeric version of the protein is not full-length and is missing the ST region residues.

While the size of this construct is relatively small, the traditional ¹H-¹⁵N HSQC spectra of the protein are not of high quality, most likely due to μs-ms dynamics in the CTD domain as indicated by Karamanos et al (ref 19). To overcome this, we recorded the methyl-TROSY spectrum of the protein, which is of high quality.

The methyl spectrum of this monomeric (DNAJB6 ΔST) construct indeed overlays very well with that of the DNAJB6JD-GF construct. This data is shown in Supplementary Figure 6e of the revised manuscript.

d) Previous work from the Rosenzweig lab on DNAJB1 suggested that transient interactions between JD and CTD play a role in autoinhibition release. Is that the case for DNAJB6? This needs to be addressed in the discussion.

While DNAJB6 and DNAJB1 have homologous JD-GF regions, their C-terminal domains are structurally very different. As such, it is unclear if any functional similarities should indeed exist.

Specifically, in the case of DNAJB1 the release of the inhibition requires the binding of Hsp70 to an additional site, located in CTDI. DNAJB6 lacks any domain that is structurally similar to that CTDI and therefore a different release mechanism must be employed. Strengthening this hypothesis, our NMR experiments found no interaction between the CTD of DNAJB6 and Hsp70. Potential release mechanisms are elaborated in the discussion section.

"In DNAJB1 the JD-GF inhibition is regulated through a second Hsp70-binding site, located in the CTDI domain of the chaperone⁵⁰. Such a site, however, is absent in DNAJB6, raising the question of how the JD-GF inhibition is indeed released and regulated in the WT DNAJB6 protein. Karamanos et al.¹⁹ suggested that the release could perhaps be mediated by interaction of client proteins with the client-binding domain of DNAJB6. The DNAJB6 chaperone is thought to contain two client binding domains with distinct client specificities: the amyloid binding ST-rich region^{25,31,42,44,70}, and the poorly characterized CTD, reported to interact with misfolded proteins prone to amorphous aggregation³⁶. Therefore, it may be that client binding to only one of these domains can release the inhibition. In such a case, interaction with some clients would release

the inhibition and transfer the proteins to Hsp70 chaperones, while that of others, that bind to the second domain, would not. For those clients that do not release the inhibition, DNAJB6 chaperones may instead function in an Hsp70-independent manner. The dependence of DNAJB6 on the Hsp70 chaperone machinery in the cell could thus potentially be predetermined by the type of client.”

4. All assigned chemical shifts for all mutants should be deposited in the appropriate database.

We have deposited all the assignments for the mutants to the BMRB

5. Figure 6 legend, parts a and b seem mixed up.

We thank the reviewer for pointing this out and have corrected the error.

Error bars in panels e and g?

The graphs in Figure 6e and g, were analyzed in a contingency table (2 by 2) and used the standard Fisher's exact test to calculate significance. Fisher's is used in relatively small sample sizes, as in our experiment, to examine the significance of the association (contingency) between two kinds of classification (normal and ruffled/undulated muscle fibers). Three tables were created (WT versus DNAJB6^{Open}, WT versus DNAJB6^{Open-QPN}, and DNAJB6^{Open} versus DNAJB6^{Open-QPN}) and then corrected for multiple comparisons using Dunn's multiple comparison test. Since the data represents a fraction of the animals exhibiting the phenotype (in percentages), and thus shows precise counts, error bars are not displayed.

6. Some referencing issues throughout.

We have gone over the paper and corrected all identified errors in referencing.

Reviewer #2 (Remarks to the Author):

The manuscript describes a set of studies that significantly advance our understanding of the mechanisms underlying mutations in DNAJB6 that are associated with a LGMD. The data is of high quality and the conclusions drawn from the data are largely convincing (but see points below).

The main novelties of the work are the findings that mutations in DNAJB6 JD and G/F region linked to LGMDD1 likely dysregulate binding of the JD to Hsp70 (mutations release resting state inhibition of the binding interaction), and that the dysregulation leads to sequestering of Hsp70s away from other roles in the cell. The G/F helix docked to the J-domain in DNAJB6 was shown previously in the structure determined by Karamanos et al., and appears to be operate analogously to what was seen in DNAJB1. The results suggest a simple correlation between helix V unbinding to the JD and binding of Hsp70 to the JD, and this correlation extends also to ATPase stimulation. It is not obvious however why some of these mutations provide more or less unbinding between helix V and the JD in the first place. In addition, a potential novelty with regard to mechanism could have been in how client proteins induce release of the J-domain inhibition (which the authors point out must be different to that of DNAJB1) but that is not addressed here.

Questions:

The mechanism by which some of the mutations affect helix V dissociation from the JD is not obvious from the residue positions and the structures shown. Some discussion of this and perhaps more analysis of the mutation sites seems appropriate. What mutation sites make direct contact with the JD, and is there any evidence for conformational flexibility not reflected in the static structures that might allow for close contacts that otherwise appear long-range in the models? Some of these substitutions are in what could be flexible loops, but have dynamics been looked at? The chemical shift perturbation data of Supp Fig 2 seems like it could be a rich source of information for understanding long range effects from the mutations.

We thank the reviewer for raising this very important point and for giving us the push needed to further structurally characterize the LGMDD1 disease mutants.

While the effect of some of the mutants can clearly be explained based on the NMR structures, the effect of other mutations, specifically those located in the disordered part of the GF, is less clear.

Based on the structures it is, for example, clear that E54A mutation breaks the salt bridge between E54, found in helix III of the JD, and R94 in the GF, destabilizing the helix III - GF interactions. Similarly, F100V mutations destabilize the aromatic-aromatic contacts made between F100, Y24 (in helix II), and F46 (in helix III). A50V and P96R potentially generate a steric clash undocking the GF from helix III of the J-domain.

We have added a description of the potential mechanisms by which these mutants may disrupt the JD-GF interaction to the text of the revised manuscript.

“The populations of the free J-domain varied greatly amongst the different LGMDD1 DNAJB6 mutants. Disease mutations located in inhibitory helix V (P96R and F100V) showed high degrees

of opening, shifting the J-domain conformational equilibrium from a completely inhibited state in the WT DNAJB6^{JD-GF} ($p_{Free} = 0$) to $p_{Free} \sim 67\%$ and $p_{Free} \sim 95\%$, respectively (Fig. 3c and supplementary Fig. 3a). The near full population shift for the mutation in residue 100 is consistent with its location in the middle of inhibitory helix V, where it normally forms aromatic-aromatic contacts with both helix II (Y24) and III (F46) of the J-domain¹⁹.”

“Interestingly, the A50V mutation also caused substantial opening of the JD-GF inhibition, shifting the equilibrium almost entirely to the open state ($p_{Free} \sim 91\%$). Residue 50 is located in helix III of the JD and, based on the NMR structure¹⁹, forms direct contacts with helix IV (Fig. 3a and supplementary Fig. 3a). It is possible that the substitution of alanine with the bulkier valine residue generates steric interference and releases the GF inhibition. Mutation to residue 54 (E54A) which forms a salt bridge with R94 in the GF region¹⁹ and thus stabilizes helix V docking, likewise induces opening, albeit to a lesser degree - increasing the population of the free JD to $\sim 43\%$ (Fig. 3c).”

The effect of the 3 mutations found in the disordered region of the GF is, however, less clear.

To gain further structural insights into the contacts made by F89, F91, and F93, we have measured a ¹⁵N-edited NOESY experiment. This experiment, in combination with the chemical shift perturbations caused by the mutations, indicates that residues F89 and F91 form contacts with residues in helix V of the J-domain, while residue F93 forms contacts with helix III. However, in order to conclusively point out the specific interactions formed between these residues a more detailed structural calculation of DNAJB6 needs to be performed and additional proton-proton NOEs need to be collected. We intend to perform these experiments, however feel that such an analysis is beyond the scope of this current paper.

The proposal is for a simple link between helix V undocking and Hsp70 binding. Flipping back and forth between the helix V docking percentages shown in Fig. 3(c) and the ATPase activities shown in Fig. 4 suggests that the correlation will be good but I suggest that a correlation plot be presented. A correlation plot provides a quantitative measure, and also reveals any nonlinearity or outliers, which might help formulate additional mechanistic hypotheses (for example, perhaps pointing to mechanisms related to the variable changes in helix 5 intensity loss noted below). The correlation with ATPase activities for full-length DNAJB6 from Fig. 6(a) should be assessed similarly.

We thank the reviewer for this excellent suggestion and have added the correlation plots to the revised manuscript.

We detect a very good correlation ($R^2=0.97$) between the fraction of the free population and the degree of Hsp70 activation by DNAJB6 JD-GF constructs. This result was added to Supplementary Figure 3 of the revised manuscript.

Good correlations between the free state population and Hsp70 binding ($R^2=0.94$) and activation ($R^2=0.98$) are also detected for the full length DNAJB6 chaperones. These correlations were added to Supplementary Fig. 8 of the revised manuscript.

The helix V intensity losses for the various mutants on binding Hsp70 are surprising and

unexpected for a simple model in which helix V undocks from the JD and provides access for Hsp70 to bind the JD. The unexpected intensity losses are noted by the authors but there aren't any real insights as to what is happening here. There is not an obvious trend across the mutants that correlates with the severity of the undocking of helix V, and a more or less rigid link between helix V and the JD seems unlikely. There could be a combination of factors, including how the mutation affects the ability of the helix V to get out of the way of Hsp70, subtle changes in pose, whether there are nonspecific contacts with the Hsp70 and perhaps chemical exchange processes, etc. If the correlation between (un)docking percentage and Hsp70 binding/ATPase activity is very high, then an in depth analysis of this is probably not very interesting, but it nonetheless begs questions about what is actually known about the Hsp70/JD complex interface and what restrictions Hsp70 binding imposes on where helix V can be positioned. If there are reliable models for any Hsp70/JD complex then perhaps this should be provided and discussed. It may be worth noting that the AlphaFold2-multimer, which produces a good quality pose with low PAE for the JD (full-length Hsp70 required), places helix V in the SBD binding site. Of course, the relevance of this is unknown but could be tested with SBD mutants.

We thank the reviewer for bringing up this point as we were also surprised by the intensity losses in helix V upon Hsp70 binding.

To test that these do not arise from a simple helix V - Hsp70 SBD interaction, we have performed two types of experiments. First we measured the binding between two mutants F93L (that shows intensity drops in helix V) and A50V (that does not) to Hsp70 in the ADP state. Both these mutants have a high percentage of free JD-GF and thus released helix V that should be available for Hsp70 binding, if such exists. Hsp70-ADP does not interact with the J-domain, but should have higher affinity for helix V if it binds it as a client. Our NMR experiments did not detect any interaction between these two mutants and Hsp70-ADP, confirming that helix V does not bind Hsp70 as a client.

Secondly, we performed binding experiments between methyl-labeled (^2H , $^{13}\text{CH}_3$ -ILVM) full length Hsp70-ATP and either DNAJB6 JD-GF mutants (F93L and A50V) or the DNAJB6 JD that lacks the GF region entirely. Identical binding profiles were observed for both the mutants and the J-domain, confirming that the GF region does not form any contacts with Hsp70.

These results of these experiments are shown in Supplementary figure 5 of the revised manuscript.

As helix V does not interact with Hsp70, the intensity reduction in this region most likely arises from the dynamics of helix docking/undocking coupled to Hsp70 binding. In this case, the intermediate exchange binding becomes the rate limiting event and therefore for the mutants in which the free helix V is the minor state the effects of that intermediate exchange binding cause the intensity drops.

For the mutants for which the free helix V is the major state, the intermediate exchange of Hsp70 binding is no longer governing the whole process and therefore no large decreases in intensity are observed.

There are some comments/concerns about the assessment of DNAJB6 anti-aggregation activity with and without Hsp70 (end of 2nd paragraph in Results, and Supp Fig. 1(b)), although it's not clear that the data here is actually necessary since the conclusion from the experiments is that "DNAJB6 chaperones can operate in an Hsp70-independent manner", which can be determined from the assays in the absence of Hsp70. In any case: Hageman & Kampinga et al. 2010 (ref 31) observed a similar result in cells (which should be cited and referred to here directly), and went further by showing that the J-domain (and so the Hsp70 interactions) of DNAJB6 (or DNAJB8) did not significantly aid in polyQ clearance for polyQ74, but did help against polyQ119. This suggests that it depends on the polyQ length. The manuscript results are for a polyQ48, which means this is not expected to be a stringent test of whether Hsp70 helps or not. Presumably any boost provided by Hsp70 would also depend on the relative concentrations, but I can't see that it is indicated what were the concentrations of either Hsp70 or DNAJB6 in the fibrilization assays of Supp. Fig 1(b). It should be noted however that the Hageman et al. results suggest that the additional activity provided by Hsp70s was due to the ability of Hsp70s to pass the polyQ to the cellular degradation machinery, which of course is not possible in this in vitro set-up. Finally, to observe a holdase activity of Hsp70, would it be necessary to include a NEF? In the canonical Hsp70 cycle, ATP hydrolysis stimulated by a J-domain induces substrate binding to the Hsp70, but release of substrate requires nucleotide exchange, which can be slow. Thus there are two possibilities that are difficult to assess here: Is it possible that Hsp70 is not turning over and becomes stalled in the absence of a NEF? Or is polyQ48 below the length and/or at a concentration below the capacity of DNAJB6? Any of these results seem possible here, and I can't see that the conclusion that Hsp70 provides no additional anti-aggregation activity is either accurate or particularly useful (or necessary for the conclusions?).

We thank the reviewer for pointing this out and have modified the statement in the text as well as added the appropriate reference by Hageman and Kampinga et al. 2010.

"Interestingly, this antiaggregation activity was not further bolstered by the addition of Hsp70 (Supplementary Fig. 1b), suggesting that DNAJB6 chaperones, in vitro, can operate in an Hsp70-independent manner. "

The authors state with regard to the data in Supp. Fig 5: "No changes were detected in other regions of the protein, strengthening our observation that client binding of DNAJB6 is not affected by the pathogenic mutations". I suggest this statement be removed since the data cited here is on a construct that is missing the ST region, which is known to be involved either directly or indirectly in substrate binding, as shown in Fig. 1(d) and 1(e). The data in Fig. 1(d) and 1(e) is a better test of whether the mutations affect substrate binding.

This sentence was corrected to "No changes were detected in other regions of the protein, strengthening our observation that the **CTD client-binding region** of DNAJB6 is not affected by the pathogenic mutations".

The work in *C. elegans* is very nice and a strong component of the manuscript. The high expression level of DNAJB6 in worm muscle cells is shown and also Meissner et al. 2011 is cited. The high DNAJB6 expression level is important for the hypothesis of Hsp70 sequestering. Is it known whether DNAJB6 is also highly expressed in human muscle cells? If it is known to be the

case, then it should be cited, and if not or it is not known, then the discussion should be modified to reflect that.

We refer to the high expression of DNAJB6 in human muscle cells in the discussion, and the appropriate references have been added:

“The high expression of DNAJB6 in muscles, compared to the brain and nervous system, combined with the decreased turnover rate and elevated levels of mutant DNAJB6^{13,35,64,75,76} (Supplementary Fig. 9e), could explain how these mutants deplete Hsp70 levels selectively in the muscle tissue.”

More minor comments:

Nomenclature of constructs is often confusing:

- It should be stated clearly in Materials and Methods that constructs are based on human isoform b of DNAJB6.

We have added this information to the Materials and Methods section

- The use of the label "WT" is used in figures whether it is full-length or the JD-GF construct. It should be made clear in all figures and legends which construct (full-length or truncated) is being used.

We have changed the labeling in the text, figures, and figure legends to clearly indicate the construct to which we refer. Full length DNAJB6 is indicated as DNAJB6, the ST-truncated monomeric construct as DNAJB6^{mono}, and the DNAJB6 residues 1-109 containing only the JD and GF regions as DNAJB6^{JD-GF}.

- It seems the JD+GF construct is referred to as either DNAJB6(GF) or DNAJB6(JD-GF). There are many occurrences of both – I've assumed there is no data presented for a GF-only construct.

We have corrected this and now all constructs of residues 1-109 are consistently referred to as DNAJB6^{JD-GF} throughout the manuscript.

- Similarly, the name given to the "monomeric WT DNAJB6" should more clearly indicate that there is an internal deletion and so not wildtype throughout (e.g., DNAJB6(mon)). This same construct seems to be sometimes referred to as DNAJB6 deltaST as well, but in some cases it's not really possible to be sure of this from the way they are presented and discussed. It's appreciated that the WT is in reference to the sequence in the JD/GF region but it is confusing without having it identified as full-length vs. truncation.

We thank the reviewer for pointing this out and now provide a clear indication in the text and materials and methods that the monomeric version of the protein is not full-length and is missing the ST region residues.

There are quite a few errors/omissions/typos:

- Fig 1(d) and 1(e), it should be indicated what are the DNAJB6 and Q48/TDP-43 concentrations used in these $t_{1/2}$ measurements.

We apologize for mistakenly omitting this information, and have added this information to the figure legends of Figure 1 of the revised manuscript.

“(d) The effect of 0.8 μ M DNAJB6 WT and disease mutants on HTT_{Ex1}-Q48 (10 μ M) aggregation half times. Data represents mean values \pm s.d (n=4). (e) The effect of 5 μ M DNAJB6 WT and disease mutants on 10 μ M TDP-43 aggregation half times. Data represents mean values \pm s.d (n=4).”

- The coordinates for the structures shown in Fig 1(b) and Fig 3(a) appears to come from 6U3R (or 6U3S), rather than the 7JSQ indicated in the legends.

We corrected this mistake.

- Fig 3(a), would be good to have helices numbered to help with following the text. In addition, the location of the HPD motif that is primarily responsible for Hsp70 interactions should be shown.

Helix numbering and the location of the HDP and now clearly indicated in the revised figure.

- In Assignments section of Materials & Methods: what sites are being referred to when indicating the "unambiguous assignment" of the constructs? Is this backbone amide or does this include backbone heavy atom or other protons, etc.?

Unambiguous assignments refer to all C $_{\alpha}$, C $_{\beta}$, CO, N, and H_N chemical shifts for which NMR assignments were obtained.

- Fig. 5(a) & 5(b), and Supp Fig. 5(a): Is the text embedded in the spectra indicating the ¹³C methyl labelling correct? This seems to be the only mention of an "LV" labelled sample, whereas all the other methyl-labelled samples are "ILVM".

We have corrected this typo and the spectrum is now labeled as an ILVM-labeled sample

- Fig. 1(a), the DNAJB6 protein should be 241 amino acids long, not 240?

Based on the Uniprot database, isoform b of the human DNAJB6 is 241 amino acids long.

- Fig. 5 legend 2nd to last sentence: "while the A50V mutant enhances activity 6.5" presumably should end with "...6.7-fold"?

The word fold was omitted by mistake. The figure legend was corrected to “(e) Steady state ATPase activity of Hsp70 alone (white) and upon incubation with monomeric WT DNAJB6 or A50V disease mutant. The WT shows no activation of Hsp70 ATPase activity, while the A50V mutant enhances the activity 6.7-fold. Data are means \pm SEM ($n = 3$).”

- Fig. 6 legend, (a) and (b) descriptions appear to be swapped.

This mistake was corrected.

- Fig. 6(e) and 6(g), should have error bars.

The graphs in Figure 6e and g, were analyzed in a contingency table (2 by 2) and used the standard Fisher's exact test to calculate significance. Fisher's is used in relatively small sample sizes, as in our experiment, to examine the significance of the association (contingency) between two kinds of classification (normal and ruffled/undulated muscle fibers). Three tables were created (WT versus DNAJB6^{Open}, WT versus DNAJB6^{Open-QPN}, and DNAJB6^{Open} versus DNAJB6^{Open-QPN}) and then corrected for multiple comparisons using Dunn's multiple comparison test. Since the data represents a fraction of the animals exhibiting the phenotype (in percentages), and thus shows precise counts, error bars are not displayed.

- Residue/methyl numbering should be indicated in Figs. 5(c) & 5(d).

Methyl numbering was added to these figures.

- Residue/methyl numbering should be indicated in Supp. Figs. 4(f) & 4(g).

Methyl numbering was added to these figures.

- Discussion, 2nd to last paragraph, the call (DNAJB6 expression levels) should be to Supp Fig 6e.

The figure referencing was corrected.

Reviewer #3 (Remarks to the Author):

The authors nicely characterize the effects of common mutations causing LGMDD1 on the structure and two molecular functions of the mutated protein DNAB6 in vitro. In doing so they provide a unifying theory of the commonality of the disease causing mutations, specifically that they impact HSP70 function in a dominant interfering fashion. The dominant fashion is particularly important because this is an autosomal dominant disease and therefore loss of DNAB6 function would not be expected to produce a dominant disease (indeed knockdown the *C. elegans* orthologue appears to produce a sterile or lethal phenotype and the mouse knockout is also embryonic lethal). Having nicely demonstrated a unifying theory for how the mutations might act at the molecular level in vitro, the authors introduce one of these mutations into *C. elegans* and find that it does indeed produce dystrophic muscles (note they did not demonstrate that it does so in a dominant fashion). The authors are successful at partially rescuing the dystrophic phenotype by introducing a second site mutation in DNAB6/DNJ-24, suggesting that at least part of the mechanism underlying the dystrophic phenotype is due to the interaction with HSP70. The authors further provide evidence for the possible HSP70 sequestration mechanism by overexpressing HSP70/HSP-1 in the presence of the mutant form of DNAB6/DNJ-24.

We thank the reviewer for these positive comments.

- What are the noteworthy results?

The most noteworthy results are: a) demonstration of a dominant mechanism of action underlying a dominant disease (this is important as much of the field has been focused on loss of function mechanisms); b) demonstration of the feasibility to develop a rapid model (*C. elegans*) for increasing mechanistic understanding and treatment of LGMDD1.

- Will the work be of significance to the field and related fields? How does it compare to the established literature? If the work is not original, please provide relevant references.

The work is highly significant to the field as it is paradigm shifting. The work will also be of interest to those studying proteostasis (including diseases), muscle physiology, and loss of muscle homeostasis with age. Compared to past literature this work is focused on incorporating personalized patient data into mechanistic understanding. Such work has been useful particularly for CFTR and is increasingly being looked at in rare diseases such as LGMDD1.

- Does the work support the conclusions and claims, or is additional evidence needed?

The work broadly supports the conclusions and claims. Unsurprisingly, the main areas for potential improvement relate to the in vivo experiments which are “proof of principle” at the end of the manuscript. For example, only phalloidin staining is used to look at actin structure, otherwise the muscle histology/physiology is not examined. To be fair, this would be a paper in and of itself (movement (and effect of interventions), other structure w/wo interventions (myosin, Z lines, M lines, mitochondria), Ca⁺⁺ levels w/wo interventions (see <https://pubmed.ncbi.nlm.nih.gov/36935171/>), PolyQ/other aggregation w/wo interventions, and possibly EM). Thus, I would encourage the authors to consider these experiments and to publish a follow on paper or papers having apparently established a novel model for LGMDD1 that could then be exploited for therapeutic discovery (and more mechanistic studies).

We thank the reviewer for these excellent suggestions and indeed intend to carry out these experiments in a follow-up study.

If I were to ask for additional experiments, it would probably just be movement data to go with Figure 6d and f (e.g. thrash assays in these 4 conditions) assuming these worms are currently in the lab this would be 1-2 weeks of work.

As per the reviewer's suggestion, we performed a thrashing assay on wild-type and DNAJB6 "open" mutant animals (panel A), as well as used a multi-worm tracker to record multiple locomotion-related parameters (Panel B). No significant differences were observed between the two groups. So while we do see clear cellular defects in the muscle integrity, these are not translated into observable locomotion defects in our worm model, most likely due to the differences in muscle structure and physiology between *C. elegans* and humans.

Other thoughts:

1) It might be reasonable to suggest that other proteins could also be binding to DNAJB6 and therefore not everything has to be dependent upon HSP70 (even if it is likely the predominant mechanism).

We believe that in the case that the disease phenotype were caused by other proteins binding to DNAJB6, it would not be suppressed by Hsp70 overexpression or DNAJB6 mutations that only abolish Hsp70 binding.

2) It might be reasonable to consider clinical data (for example <https://pubmed.ncbi.nlm.nih.gov/36427278/>) vs. your unifying model in the discussion section (e.g. the model doesn't immediately fit the clinical data, of course this could be due to individual genomic differences as well)

We thank the reviewer for bringing this up and we have looked at the clinical data. We, however, do not see a clear correlation between the disease onset and the degree of JD-GF opening. Generally, the more open mutants do show earlier onset ages, however mutations A50V and F91L/I are an exception. F91 mutants are more severe, despite having a relatively

low percentage of open population, and A50V is a late onset mutation despite being almost completely open. It is however an important point, and we intend to test in future studies the possibility that additional factors can contribute to the disease, especially in the case of the F91 mutation.

- Are there any flaws in the data analysis, interpretation and conclusions? - Do these prohibit publication or require revision?

The main reservations I have are with potentially overstating things. For example “unregulated” vs. “dysregulated” and “Excitingly” and “conclusively” and “disease phenotype” vs. potentially multiple phenotypes with potentially multiple mechanisms.

We thank the reviewer for pointing this out. We have removed all statements of “conclusively” and have limited the use of “excitingly” to once in the entire text.

As the term “dysregulated” carries other potential implications, we would, however, prefer to continue to use “unregulated”.

The second reservation I have is with suggesting the protein aggregation is not altered in vivo (I think this is a wording issue confusing DNAJB6 direct chaperone activity vs. direct HSP70 chaperone activity), esp. as it is part of the clinical phenotype (and probably is occurring based upon hsp-1 knockdown results in wormbase <https://pubmed.ncbi.nlm.nih.gov/15084750/>). I raise this as the phenotype of hsp-1 includes increased protein aggregation (and a movement defect and altered mitochondrial structure) amongst other things (for example probably altered stress responsiveness based upon impacts on daf-16 localization). Additionally, you’ve not assessed protein aggregation in vivo thus it might be better to tone down the language around this a bit.

We have revised the text to indicate that the anti-aggregation activity of DNAJB6 LGMDD1 mutants is not affected *in vitro*.

We also completely agree that the effect observed in patients and in cells is not due to the alteration of the direct DNAJB6 anti-aggregation activity, but rather an effect on the activity of the Hsp70 chaperone system that eventually causes the accumulation of protein aggregates. We have revised the text in the discussion to indicate it more clearly.

“Our findings indicate that this high-affinity, non-productive interaction with mutant DNAJB6 depletes cellular levels of Hsp70 chaperone, disrupting protein homeostasis and leading to protein aggregation. Therefore, it is not, as previously suggested, the loss-of-function of DNAJB6 that underlies the disease, but rather the unregulated binding of DNAJB6 LGMDD1 mutants to Hsp70.”

Lastly, the images of Open in 6 d vs f display quite distinct pathologies. D is reminiscent of DMD (<https://pubmed.ncbi.nlm.nih.gov/11696327/>) whereas F is reminiscent of tropomodulin mutants/nemaline pathologies (<https://pubmed.ncbi.nlm.nih.gov/17976644/>) which present more severely and are associated with protein aggregation and defects more at the myotendonous

junction than the Z/M line. It might be worth either subclassifying minor vs. severe (in e and g graphs) or displaying consistent minor images in d and f.

We revised the figure according to the reviewer's suggestion, displaying consistent minor phenotypes in d and f.

- Is the methodology sound? Does the work meet the expected standards in your field?
Yes and yes.

- Is there enough detail provided in the methods for the work to be reproduced?

Yes.

REVIEWERS' COMMENTS

Reviewer #1 (Remarks to the Author):

The revised manuscript is much improved and most of my initial points are dealt with. A few concerns remain:

1. *C. elegans* model

I get the value of an artificially released mutant and how it adds to the proposed model. However, it naturally raises questions about the effect of the artificial substitutions to the structure and perhaps function of the protein. Figure S10 is using methyl NMR spectra to probe structural differences in DNAJB6-open but I note that methyl groups are absent from large parts of the GF region. In general, I still believe that data on A50V or F100V would be a valuable addition to the paper.

2. NMR studies

a) Although the calculation of fractional populations is now nicely described in the revised manuscript, my comment about toning down the language regarding release of autoinhibition is not addressed. From reading the revised manuscript it is clear that the effect of mutations in the folded parts of JD is well understood. However, that cannot be said for the mutations in the disordered parts of the GF. If the authors are not willing to follow my initial suggestion, I will insist on a much clearer distinction between the mutants in the disordered and folded regions of JD.

b) It would really be helpful to change DNAJB6-mono to DNAJB6- Δ ST to be consistent with the literature and avoid confusing statements such as full-length monomeric DNAJB6. I am not convinced about the argument of poor quality $^1\text{H} - ^{15}\text{N}$ spectra as ref 19 (that is being used throughout the manuscript) is largely based on ^{15}N experiments on DNAJB6- Δ ST as far as I can tell.

Reviewer #2 (Remarks to the Author):

The text manuscript has improved clarity and better embeds the current study in the context of what was previously known. The new experiments aimed at revealing molecular mechanisms are also very welcome. While the new experiments are not entirely conclusive, they help better define the unknowns.

It is useful to have included in the text a brief discussion of where the mutations are relative to what is known structurally about the interactions between helix V and the JD, and I agree that it is fine to simply indicate in this manuscript the mutation sites that cannot yet be rationalized mechanistically.

The additional NMR experiments on F93L and A50V in the presence of Hsp70 are useful; while they do not explain the intensity differences in binding to Hsp70, they suggest that binding to the SBD is unlikely to be the source of the unexplained intensity changes.

Remaining questions/concerns -

The modified form of the sentence at the end of the paragraph describing the Hsp70-independent activity of DNAJB6 is not entirely satisfactory (lines 104-107). Why is this particularly interesting given that this is what has been shown previously, and the fact that Q48 is not a stringent test for this in any case? In addition, the concerns about the absence of a NEF in these assays in vitro are not addressed.

Fig.1a and Supp Figs. 6a and 6b still indicate the terminal residue is number 240, rather than 241.

Why are only 6 methyl residue numbers indicated in Figs. 5c and 5d? Also, full residue numbers should be provided for the methyl CSPs shown in Supp. Figs. 6f and 6g, Supp. Figs. 7b,7c and 7d, and Supp. Figs. 10d and 10e. The residue numbers for the amide backbone CSPs in Supp. Fig. 2. should also be shown.

Finally, I share concerns about over-interpreting the in vivo work and suggest that additional caution should be used in discussing the in vivo work, especially in light of new data provided in response to Reviewer 3 that the mutation has no effect on worm motility. Probably, the thrashing data should be included in the manuscript as part of an expanded discussion of potential limitations of the interpretation directly linking helix V/JD dissociation with disease pathology.

Reviewer #3 (Remarks to the Author):

The authors have considered my comments and revised accordingly. I feel they have adequately addressed my comments and have no further actions for them.

Reviewer #1 (Remarks to the Author):

The revised manuscript is much improved and most of my initial points are dealt with. A few concerns remain:

We thank the reviewer for the positive comments.

1. *C. elegans* model

I get the value of an artificially released mutant and how it adds to the proposed model. However, it naturally raises questions about the effect of the artificial substitutions to the structure and perhaps function of the protein. Figure S10 is using methyl NMR spectra to probe structural differences in DNAJB6-open but I note that methyl groups are absent from large parts of the GF region. In general, I still believe that data on A50V or F100V would be a valuable addition to the paper.

We agree with the reviewer and therefore show in the manuscript a detailed comparison between our artificially open mutant and disease mutants with high population of the undocked (free) JD-GF conformation, such as A50V.

First, we compared the methyl spectra of the two DNAJB6 variants and found that the mutant is very similar to DNAJB6 A50V, aside from residues in close proximity to the mutation sites. As NMR chemical shifts are very sensitive to structural changes in the protein, we can conclude that the DNAJB6 open mutant is indeed structurally similar to the disease mutants.

Second, we have performed several experiments to validate that the mutations we introduced do not affect the function of the chaperone in aggregation prevention or as an Hsp70 co-chaperone. Lastly, we tested that the artificially open mutant activates the Hsp70 machinery in an unregulated manner, similarly to the disease mutants.

The results of these experiments are shown in supplementary figure 10 of the manuscript.

The introduction of A50A and F100V mutations into *C. elegans* by CRISPR, however, is a lengthy and labor intensive process, and we feel it is therefore outside the scope of this current manuscript.

2. NMR studies

a) Although the calculation of fractional populations is now nicely described in the revised manuscript, my comment about toning down the language regarding release of autoinhibition is not addressed. From reading the revised manuscript it is clear that the effect of mutations in the folded parts of JD is well understood. However, that cannot be said for the mutations in the disordered parts of the GF. If the authors are not willing to follow my initial suggestion, I will insist on a much clearer distinction between the mutants in the disordered and folded regions of JD.

We agree with the reviewer that structural effects of the mutations in the disordered part of the GF that lead to the undocking of helix 5 are not well understood. We therefore, as suggested by the reviewer, revised the text in the manuscript to clearly convey this point –

“Mutations located in the disordered part of the GF had smaller degrees of JD-GF destabilization, with p_{Free} of 18% in F91I and 37% in F93L disease mutants. Mutation to residue 89 (F89I), which only forms contacts with helix IV of the JD, led to a very minor shift from the inhibited state, with just 6% free conformation detected (Fig. 3c and supplementary Fig. 3a). More detailed structural studies, however, are required to provide a mechanistic understanding as to how these mutations induce helix V undocking from the J-domain.”

However it is important to note that since all the mutations (including those in the disordered GF region) display chemical shifts that titrate in a linear fashion, with the endpoints being the fully inhibited WT DNAJB6^{JD-GF} on one end and the free DNAJB6^{JD} on the other, we can conclude that they all undergo the same two-state interconversion between inhibited/free conformations of the DNAJB6^{JD-GF} associated with the release of the autoinhibitory helix V.

b) It would really be helpful to change DNAJB6-mono to DNAJB6- Δ ST to be consistent with the literature and avoid confusing statements such as full-length monomeric DNAJB6.

We have replaced the confusing statements and now consistently refer to this variant as monomeric DNAJB6 (DNAJB6 Δ ST).

I am not convinced about the argument of poor quality ¹H – ¹⁵N spectra as ref 19 (that is being used throughout the manuscript) is largely based on ¹⁵N experiments on DNAJB6- Δ ST as far as I can tell.

As the methyl spectrum of DNAJB6 Δ ST and disease mutants was of higher quality compared to the ¹H-¹⁵N TROSY-HSQC spectra of ²H/¹⁵N-labeled proteins, we decided to use that throughout the manuscript to characterize the monomeric DNAJB6 constructs. This methyl spectrum of DNAJB6 Δ ST monomeric constructs overlaid very well with that of the DNAJB6^{JD-GF} construct.

Reviewer #2 (Remarks to the Author):

The text manuscript has improved clarity and better embeds the current study in the context of what was previously known. The new experiments aimed at revealing molecular mechanisms are also very welcome. While the new experiments are not entirely conclusive, they help better define the unknowns.

1) It is useful to have included in the text a brief discussion of where the mutations are relative to what is known structurally about the interactions between helix V and the JD, and I agree that it is fine to simply indicate in this manuscript the mutation sites that cannot yet be rationalized mechanistically.

We have revised the relevant section in the text to clearly indicate for which mutation sites the release of helix V can be explained mechanistically, and for which further structural information is required.

“Mutations located in the disordered part of the GF had smaller degrees of JD-GF destabilization, with p_{Free} of 18% in F91I and 37% in F93L disease mutants. Mutation to residue 89 (F89I), which only forms contacts with helix IV of the JD, led to a very minor shift from the inhibited state, with just 6% free conformation detected (Fig. 3c and supplementary Fig. 3a). More detailed structural studies, however, are required to provide a mechanistic understanding as to how these mutations induce helix V undocking from the J-domain.

Interestingly, the A50V mutation also caused substantial opening of the JD-GF inhibition, shifting the equilibrium almost entirely to the open state ($p_{Free} \sim 91\%$). Residue 50 is located in helix III of the JD and, based on the NMR structure¹⁹, forms direct contacts with helix IV (Fig. 3a and supplementary Fig. 3a). It is possible that the substitution of alanine with the bulkier valine residue generates steric interference and releases the GF inhibition. Mutation to residue 54 (E54A) which forms a salt bridge with R94 in the GF region¹⁹ and thus stabilizes helix V docking, likewise induces opening, albeit to a lesser degree - increasing the population of the free JD to $\sim 43\%$ (Fig. 3c).”

2) The additional NMR experiments on F93L and A50V in the presence of Hsp70 are useful; while they do not explain the intensity differences in binding to Hsp70, they suggest that binding to the SBD is unlikely to be the source of the unexplained intensity changes.

In addition to the experiments that show that Hsp70 SBD does not interact with DNAJB6 JD-GF mutants, we also performed binding experiments between methyl-labeled (²H, ¹³CH₃-ILVM) full length Hsp70-ATP and either DNAJB6 JD-GF mutants (F93L and A50V) or the DNAJB6 JD that lacks the GF region entirely. We obtained identical binding profiles for both the mutants and the J-domain, which show that the GF region and helix V do not form any direct contacts with Hsp70.

Based on these results, we suggest that the intensity reduction in helix V most likely arises from the dynamics of helix docking/undocking coupled to Hsp70 binding. In this case, the intermediate exchange binding becomes the rate limiting event and therefore, for the mutants in which the free helix V is the minor state, the effects of that intermediate exchange binding cause the intensity drops. For the mutants for which the free helix V is the major state, the intermediate exchange of Hsp70 binding is no longer governing the whole process, and therefore no large decreases in intensity are observed.

Remaining questions/concerns -

3) The modified form of the sentence at the end of the paragraph describing the Hsp70-independent activity of DNAJB6 is not entirely satisfactory (lines 104-107). Why is this particularly interesting given that this is what has been shown previously, and the fact that Q48 is not a stringent test for this in any case? In addition, the concerns about the absence of a NEF in these assays in vitro are not addressed.

We have revised the sentence to clearly indicate that this result was shown *in vitro* for Q48. In addition, we also removed the word “interestingly” from the beginning of the sentence.

“This antiaggregation activity for HTT_{Ex1}-Q48 was not further bolstered by the addition of Hsp70 (Supplementary Fig. 1b), suggesting that DNAJB6 chaperones, in vitro, can operate in an Hsp70-independent manner.”

As DNAJB6 dependence on Hsp70 chaperones was previously only tested in cell lines, we feel that it is important to show our *in vitro* result, which is independent of any additional factors.

Regarding the NEF – *In vitro* aggregation-prevention assays that rely on the “holdase” activity of Hsp70 do not require NEF addition (<https://doi.org/10.1073/pnas.140202897>, <https://doi.org/10.1016/j.bpj.2020.12.019>, <https://doi.org/10.1021/acscchembio.7b01039>).

4) Fig.1a and Supp Figs. 6a and 6b still indicate the terminal residue is number 240, rather than 241.

We thank the review for pointing this out and have corrected this mistake. The figures now indicate 241 as the terminal residue.

5) Why are only 6 methyl residue numbers indicated in Figs. 5c and 5d? Also, full residue numbers should be provided for the methyl CSPs shown in Supp. Figs. 6f and 6g, Supp. Figs. 7b,7c and 7d, and Supp. Figs. 10d and 10e. The residue numbers for the amide backbone CSPs in Supp. Fig. 2. should also be shown.

The numbering shown in Figure 5c and 5d represent the domain boundaries and not methyl residue numbering. We have revised the figure to make this more clear. As methyl labeling is sparse, we feel that the current representation is more informative than indicating the exact residue numbering in these figures.

As requested, we have added the numbering to Supp. Fig. 2, Supp. Figs. 6f and 6g, Supp. Figs. 7c and d, as well as clearly indicate the domain boundaries for Supp. fig. 7b, and Supp. figs. 10d and 10e.

6) Finally, I share concerns about over-interpreting the *in vivo* work and suggest that additional caution should be used in discussing the *in vivo* work, especially in light of new data provided in response to Reviewer 3 that the mutation has no effect on worm motility. Probably, the thrashing data should be included in the manuscript as part of an expanded discussion of potential limitations of the interpretation directly linking helix V/JD dissociation with disease pathology.

As suggested by the reviewer, we have added the result of the thrashing experiment to the manuscript and as supplementary figure 9e.

“It is important to note that no gross locomotion defects were detected in mutant worms in a thrashing assay (Supplementary Fig. 9e), indicating that unlike in human patients, structural defects in muscle integrity are not sufficient to disrupt mobility in C. elegans.”

In addition, the potential limitations of associating our findings directly with disease pathology are discussed in the “discussion section” of the revised manuscript.

“We, however, did not detect observable locomotion defects in mutant C. elegans animals, suggesting that in this model organism, the structural defects in muscle integrity are not sufficient to disrupt mobility. This is most likely due to the differences in muscle structure and physiology between worms and humans⁷⁵. Testing the effect of LGMDD1 mutations in additional model organisms is therefore needed to shed light on the complex pathological implications of these mutations.”

Reviewer #3 (Remarks to the Author):

The authors have considered my comments and revised accordingly. I feel they have adequately addressed my comments and have no further actions for them.

We thank the reviewer for the positive feedback.